The contribution of Nintendo Wii Fit series in the field of health: a systematic review and meta-analysis

http://orcid.org/0000-0002-5669-5337 Tripette Julien 1 2 tripette.julien@ocha.ac.jp
Murakami Haruka 2
Ryan Katie Rose 2
Ohta Yuji 1
Miyachi Motohiko 2
1 Ochanomizu University , Bunkyo, Tokyo , Japan
2 Department of Physical Activity Research, National Institutes of Biomedical innovation, Health and Nutrition , Shinjuku, Tokyo , Japan
Keogh Justin
Electronic publication date: 2017 Sep 5
Publication date: 2017
Volume: 5
Electronic Location ID: e3600
Received 2017 Apr 4; Accepted 2017 Jun 29
Copyright: © 2017 Tripette et al.
Copyright year: 2017
Copyright holder: Tripette et al.
License: This is an open access article distributed under the terms of the Creative Commons Attribution License, which permits unrestricted use, distribution, reproduction and adaptation in any medium and for any purpose provided that it is properly attributed. For attribution, the original author(s), title, publication source (PeerJ) and either DOI or URL of the article must be cited.
License URL: https://creativecommons.org/licenses/by/4.0/

Keywords: Wii Fit, Balance training, Health and fitness, Health promotion, Active video games, Rehabilitation, Prevention of chronic diseases

Funding: Fonds de recherche du Québec—Santé and the Japan Society for the Promotion of Science This work was supported by the Fonds de recherche du Québec—Santé and the Japan Society for the Promotion of Science. The funders had no role in study design, data collection and analysis, decision to publish, or preparation of the manuscript.

==============================
Background

Wii Fit was originally designed as a health and fitness interactive training experience for the general public. There are, however, many examples of Wii Fit being utilized in clinical settings. This article aims to identify the contribution of Wii Fit in the field of health promotion and rehabilitation by: (1) identifying the health-related domains for which the Wii Fit series has been tested, (2) clarifying the effect of Wii Fit in those identified health-related domains and (3) quantifying this effect.

Method

A systematic literature review was undertaken. The MEDLINE database and Games for Health Journal published content were explored using the search term “Wii-Fit.” Occurrences resulting from manual searches on Google and material suggested by experts in the field were also considered. Included articles were required to have measurements from Wii Fit activities for at least one relevant health indicator. The effect of Wii Fit interventions was assessed using meta-analyses for the following outcomes: activity-specific balance confidence score, Berg balance score (BBC) and time-up-and-go test (TUG).

Findings

A total of 115 articles highlighted that the Wii Fit has been tested in numerous healthy and pathological populations. Out of these, only a few intervention studies have focused on the prevention of chronic diseases. A large proportion of the studies focus on balance training (N = 55). This systematic review highlights several potential benefits of Wii Fit interventions and these positive observations are supported by meta-analyses data (N = 25). For example, the BBC and the TUG respond to a similar extend to Wii Fit interventions compared with traditional training.

Conclusion

Wii Fit has the potential to be used as a rehabilitation tool in different clinical situations. However, the current literature includes relatively few randomized controlled trials in each population. Further research is therefore required.

Introduction

The past decade saw the emergence of home-based active video games (AVG), with the Wii (Nintendo Co. Ltd., Kyoto, Japan) being released in 2006, followed by the PlayStation Move (Sony Corp, Tokyo, Japan) and the Kinect (Microsoft, Redmond, WA, USA) in 2010. These systems take advantage of accelerometry and video camera-mediated motion detection technologies to track the player’s movements and convert them into gaming commands. The Wii offers an original game modality with the Wii Balance Board accessory, which can be used as a weighing scale or as a gamepad sensitive to body sway (Clark et al., 2010).

Among the home-based AVG, the well-known Wii Fit series (Nintendo, Japan) runs on the Wii console and consists of a combination of both serious and entertaining activities requiring body movement to fulfill gaming commands. The software displays various kinds of health metrics (body mass index, number of kilocalories burned over a given period) encouraging the players to improve their physical fitness. Whilst the Wii Fit was primarily designed to be used in homes by healthy individuals for health and fitness purposes, an overview of the literature indicates that physical therapists and physicians from different medical fields include the use of Wii Fit in their clinical practice. For instance, the National Stroke Audit: Rehabilitation Services Report recently indicated that 76% of Australian hospitals have a Wii console available to aid with the rehabilitation of stroke patients (the National Stroke Foundation, 2012 in Levac et al., 2010).

Many reviews have focused on AVG and their effects on health and describe mitigated outcomes (LeBlanc et al., 2013; Peng, Crouse & Lin, 2013). However, the distinction between Wii Fit and other AVG was not always clear, resulting in the inability to ascertain an objective picture of the contribution from the Wii Fit. The goals for this systematic review are as follows: Goal 1: Identifying the health-related domains (i.e., populations and clinical situations) in which the Wii Fit series has already been tested or used. A scientific database search with reasoned exclusion criteria was undertaken.

Goal 2: Understanding the effect of Wii Fit in the identified populations (cf. Goal 1). A qualitative systematic review of studies including Wii Fit interventions was performed, with particular attention given to health and physical activity outcomes.

Goal 3: When possible, quantification of the effect Wii Fit has on selected health-related domains was achieved by conducting meta-analyses.

Methods

Literature search

The selection process is summarized in a PRISMA flow diagram (Fig. 1). Several strategies were adopted: (1) The MEDLINE database was used to conduct a systematic search using the following keywords: “wii fit,” “wii-fit” and “wiifit” (occurrences: N = 122). (2) The same keywords were used to search for additional articles in the Mary Ann Liebert, Inc. Games and Health Journal (N = 121). (3) Additional peer-reviewed articles were identified during manual searches via Google Search (Google Inc., Mountain View, CA, USA) (N = 1). (4) Articles suggested by authors active in the field of AVG (N = 10) or identified in the reference section of eligible papers (N = 46). (5) Only papers in English, French or Japanese were eligible for this review. The search and data extraction were performed by two independent researchers (Murakami H and Tripette J) and any discrepancies were resolved by a third contributor (Miyachi M).

Figure 1 Flow diagram for the selection of studies included in the systematic review and the meta-analyses.

Details about exclusion criteria and the selection process can be found in Table 1 and the “Methods.”

The literature search was completed in June 2015. A total of 200 articles were identified. In order to meet the primary inclusion criteria, studies were required to: (1) have a primary focus on any software of the Wii Fit series, and (2) focus on a recognized health issue. A total of 279 articles were screened after the identification and removal of 21 duplicates (Fig. 1).

Goal 1: identification of health domains

The exclusion criteria applied to identify medical domains in which the Wii Fit has already been tested or used are described in Table 1. The identification process involved screening titles and abstracts. The full texts were read when the abstracts provided insufficient details (Fig. 1 and Table 1). The results are shown in Table 2.

Table 1 Summary of exclusion criteria.

Literature review stage	Exclusion criteria	
Screening	The study was not about any Wii Fit software

The study does not focus on any health issue

The article does not describe an original study

The article was one of the following: letter, commentaries, symposium reports, interviews, conference abstracts, study protocols, reviews

The article has not been peer-reviewed

	
Goal 1	Wii Fit was not the main component of the intervention

Wii Fit was used only to induce a stimulus without being the main object of the study

The study focused on the Wii Fit avatar system rather than on its gaming content

The article reports the development of software for the Wii Balance Board

The study focuses on the Wii Balance Board capabilities not on the Wii Fit gaming content

	
Goal 2	The article does not describe an intervention study

The study does not include an objective assessment for at least one health or physical activity indicator, assessed quantitatively

The protocol includes less than five subjects or does not report average or median values

	
Goal 3	The study does not include numerical data for the activities-specific balance confidence test (ABC), Berg balance score (BBS), or the time-up-and-go test (TUG)1

The data reported for ABC, BBS and TUG were not mean ± SD, or does not allow the calculation of a mean values and the imputation of SD2

The magnitude of changes in ABC, BBS or TUG, were expected to be important due to patients’ initial condition and regardless of the chosen rehabilitation program (e.g. post-surgery orthopedic patients)3

The population sample’s average age was less than five-years old4

	
Notes:

SD, standard deviation.

1 Examples of excluded studies for this selection criteria are (Morone et al., 2014; Wall et al., 2015).

2 Examples of excluded studies for this selection criteria are (Bieryla & Dold, 2013; Janssen, Tange & Arends, 2013; Rendon et al., 2012; Bainbridge et al., 2011).

3 An example of an excluded study for this selection criteria is (Fung et al., 2012).

4 An example of an excluded study for this selection criteria is (Salem et al., 2012).

Table 2 Wii Fit studies, health domains and populations of interest.

Juvenile population	
Healthy children/adolescents1 (Levac et al., 2010; Graves et al., 2010;2 Owens et al., 2011;2 White, Schofield & Kilding, 2011; O’Donovan, Roche & Hussey, 20142)

Overweight children/adolescents (Owens et al., 2011; O’Donovan, Roche & Hussey, 2014)2

Children with developmental delay (Salem et al., 2012; Ferguson et al., 2013; Hammond et al., 2014; Mombarg, Jelsma & Hartman, 2013; Jelsma et al., 2014)

Children with migraine (Esposito et al., 2013)

Children with Raynaud disease (Qualls et al., 2013)

Children with cystic fibrosis (O’Donovan et al., 2014; del Corral et al., 2014)

Children with cerebral palsy (Ramstrand & Lygnegård, 2012; Jelsma et al., 2013; Tarakci et al., 2013; Ballaz et al., 2014)

Adolescents with autism spectrum disorders (Getchell et al., 2012)

	
Young adults and middle-age adults	
Healthy adults1,3 (Gras, Hummer & Hine, 2009; Graves et al., 2010;2 Owens et al., 2011;2 Miyachi et al., 2010; Deutsch et al., 2011; Gioftsidou et al., 2013; Melong & Keats, 2013; Douris et al., 2012; Garn et al., 2012;2 Lyons et al., 2012;2 Michalski et al., 2012; O’Donovan & Hussey, 2012; Griffin et al., 2013; Khan, Parvaiz & Vassallo, 2012; Tietäväinen et al., 2013; Lee, Lee & Park, 2014; Monteiro-Junior et al., 2014; Park, Lee & Lee, 2014; Tripette et al., 2014a; Xian et al., 2014; Cone, Levy & Goble, 2015; Naumann et al., 2015)

Healthy women (Cummings & Duncan, 2010; Nitz et al., 2010; Jacobs et al., 2011;2 Worley, Rogers & Kraemer, 2011; Steenstrup et al., 2014;4 Tripette et al., 2014b)

Overweight adults (Owens et al., 2011;2 Garn et al., 2012;2 Lyons et al., 2012;2 Jacobs et al., 2011;2 Guderian et al., 2010; Mullins et al., 2012)

Depressed soldiers (Reger et al., 2012)

Adults with drug dependency (Cutter et al., 2014)

Women with systemic lupus erythematosus (Yuen et al., 2011, 2013)

Adults with vestibular disorders (Meldrum et al., 2015; Meldrum et al., 20125)

Patients in orthopedic rehabilitation (Fung et al., 2012; Baltaci et al., 2013; Wikstrom, 2012; Sims et al., 2013; Punt et al., 2015)

Amputees (Miller et al., 2012)

COPD patients (Albores et al., 2013)

Diabetic patients (Kempf & Martin, 2013)

Hemodialysis patients (Cho & Sohng, 2014)

Lower back pain patients (Kim et al., 2014)

Adults with multiple sclerosis (Brichetto et al., 2013; Nilsagård, Forsberg & von Koch, 2013; Plow & Finlayson, 2014; Prosperini et al., 2013; Forsberg, Nilsagård & Boström, 2015; Robinson et al., 2015)

Cancer patients6 (Hoffman et al., 2013, 2014)

Stroke patients6 (Morone et al., 2014; Barcala et al., 2013; Yatar & Yildirim, 2015; Bower et al., 2014; Hung et al., 2014; Subramaniam, Wan-Ying Hui-Chan & Bhatt, 2014; Omiyale, Crowell & Madhavan, 2015)

Spinal cord injury patients (Wall et al., 2015)

	
Senior populations	
Healthy seniors (Bieryla & Dold, 2013; Janssen, Tange & Arends, 2013; Rendon et al., 2012;2 Williams et al., 2011; Yamada et al., 2011;2 Bateni, 2012; Duclos et al., 2012; Franco et al., 2012; Orsega-Smith et al., 2012; Toulotte, Toursel & Olivier, 2012;2 Chao et al., 2013; Chao et al., 2014; Cho, Hwangbo & Shin, 2014; Taylor et al., 2014; Chao et al., 2015; Nicholson et al., 2015; Roopchand-Martin et al., 2015)

Senior with balance impairment7 (Janssen, Tange & Arends, 2013; Rendon et al., 2012;2 Bainbridge et al., 2011; Daniel, 2012; Toulotte, Toursel & Olivier, 2012;2 Pigford & Andrews, 2010; Williams et al., 2010; Agmon et al., 2011; Yamada et al., 2011;2 Chan et al., 2012; Jorgensen et al., 2013)

Seniors with cognitive impairments8 (Padala, Padala & Burke, 2011; Esculier et al., 2012; dos Santos Mendes et al., 2012; Padala et al., 2012; Pompeu et al., 2012; Mhatre et al., 2013; Esculier, Vaudrin & Tremblay, 2014; Goncalves et al., 2014; Liao et al., 2015)

Seniors with peripheral neuropathy (Laver et al., 2011)

Other senior population (Hakim et al., 2015)

	
Notes:

1 Not including overweight populations.

2 Some papers focused on various populations may appear in several fields.

3 Not including studies that focus on healthy adult women only.

4 The study included healthy subjects but targeted women with urinary incontinence.

5 Patients with « other neurological disorders » were included as well.

6 Includes both middle-age adults and seniors.

7 Includes subjects referred for rehabilitation, presenting a history of accidental falls, having fear of falling or described as frail or pre-frail.

8 Includes both Parkinson’s and Alzheimer’s patients. Intervention studies eligible for inclusion in the systematic review are described in further detail in Tables 3 and 4.

Goal 2: systematic review, data extraction and synthesis

A qualitative systematic review was performed to understand the effect of Wii Fit in the previously identified health domains. This study followed the 2009 PRISMA guidelines for the conductance of systematic reviews and meta-analyses (Liberati et al., 2009) (see, Data S1). The exclusion criteria which were applied at this stage are described in Table 1.

The content of each eligible article was extracted according to the following protocol: (1) Study identification (first author’s name, year and country), (2) methodological details (study design, sample size, population characteristics, etc.), (3) activities used, (4) description of each identified primary or secondary health and physical activity outcome and (5) key findings (i.e., pre- and post-intervention as well as differences between Wii Fit and control groups) (Tables 3 and 4).

Table 3 Wii Fit interventions for health status and well-being improvement1

Authors (year), country	Population characteristics	Study design	Wii Fit activities (or other video games)	Outcomes and measures	Key findings and data used for the meta-analyses	
Healthy population	
Nitz et al. (2010), Australia	Women (N = 10)
Age range: 30–60 years
Mean age: 47 ± 10 years	Intervention
One group
Duration: 10 weeks (30 min, two sessions/week)
Location: Home (supposedly)	Not specified, possibly all the Wii Fit’s activities	Physical fitness (6 min walk test, lower limb strength), body composition, balance and functional mobility (TUG, TUGcog, step test, CTSIB, basic balance master test), well-being (home-made scale), adherence (attendance)	Improvement for some balance tests and lower limb strength. The overall attendance was 70%.
Adverse events: No
TUG (s):
Wii Fit group
Pre-intervention: 4.93 ± 0.76
Post-intervention: 5.00 ± 0.73	
Owens et al. (2011), USA	Eight families (parents and children, F/M, N = 13/8)
Age range: 8–44 years	Intervention
One group (statistical analysis: children vs. adults)
Duration: 13 weeks (no further specifications: naturalistic approach)
Location: Home	Not specified (subjects used the four categories of activities: yoga, strength, aerobics, balance)	PA (accelerometry), body composition, balance (SOT), physical fitness (VO2max, upper limb strength, flexibility), adherence (playing time)	No significant change was noted in most of the physical fitness outcomes. Peak VO2 increased in children only. Adherence declined over time. In realistic conditions Wii Fit may not provide sufficient stimulus for fitness improvement
Adverse events: No	
Tripette et al. (2014b), Japan	Postpartum women (N = 34)
Mean age: 32 ± 5 years	Intervention (RCT)
Two groups: Wii Fit vs. passive control
Duration: five weeks (30 min, daily)
Location: Home	All activities included in the Wii Fit Plus software	Body composition, physical fitness (flexibility and strength), energy intakes (questionnaire), adherence (playing time)	Women playing Wii Fit lost more weight than their passive control counterpart. They expended an average 4,700 ± 2,900 kcal playing Wii Fit and decrease their energy intakes
Adverse events: lower back pain (N = 1), ankle twist (N = 1) and wrist tendinitis (N = 1)	
Chronic diseases	
Albores et al. (2013), USA	COPD patients (F/M, N = 14/6)
Mean age: 68 ± 10 years	Intervention
One group
Duration: 12 weeks (30 min, daily)
Location: Home	Aerobics: Basic run, free step; Training plus: Bird’s eye Bull’s-eye, obstacle course	Primary: physical fitness (ESWT and other tests)
Secondary: health status (CRQ-SR, dyspnea assessment)	Home-based Wii Fit training improved physical fitness and overall health status but not dyspnea in COPD patients
Adverse events: No	
Kempf & Martin (2013), Germany	T2DM patients (F/M, N = 119/101)
Age range: 50–75 years
Mean age: 61 ± 8 years	Intervention (RCT)
Two groups: Wii Fit-traditional care vs. traditional care-Wii Fit
Duration: 12 weeks (30 min/day)
Location: Home	Not specified (supposedly, all the activities included in Wii Fit Plus)	Primary: glycemic variations (HbA1c) and various blood markers
Secondary: body composition, blood pressure, PA (questionnaire), adherence (retention), health status (SF-12, PAID) and well-being and quality of life (WHO-5, CESD)	Subjects adhered to the Wii Fit intervention (retention rate: 80%). Playing Wii Fit on a daily basis significantly decreased HbA1c in T2DM patients (−0.3 ± 1.1). Fasting glucose, weight, BMI, PA, as well as other well-being outcomes were also improved
Adverse events: No	
Cho & Sohng (2014), Korea	Hemodialysis patients
(F/M, N = 18/28)
Mean age: 59 ± 8 years	Intervention (RCT)
Two groups: Wii Fit vs. passive control
Duration: eight weeks (30 min, three sessions/week)
Location: Hospital	Yoga: Chair, Half Moon, Standing Knee (supposedly); strength: Torso Twist, Triceps Extension (supposedly); balance: Balance Bubble, Tightrope Walk; aerobics: Basic Steps (supposedly), Hula Hoop; training plus: Big Top Juggling, Bird’s-Eye Bulls-Eye, Rhythm Kung Fu, Rhythm Parade (+5 other activities that were not explicitly named)	Physical Fitness (back strength, handgrip, leg strength, sit-and-reach, single leg stance test), body composition (bioimpedancemetry), fatigue (analogue scale)	Significant improvements were noted for physical fitness, body composition and fatigue in the Wii Fit group but not the control group, suggesting that this software could be used for health promotion program in hemodialysis patients
Adverse events: No	
Kim et al. (2014), Korea	Middle-aged women with lower back pain
(N = 30)
Mean age: 47 years	Intervention (RCT)
Two groups: Wii Fit vs. traditional therapy
Duration: four weeks (30 min, three sessions/week)
Location: not specified	Yoga: Chair, Deep Breathing, Half Moon, Palm Tree, Sun Salutation (supposedly), Tree, Warrior	Pain (visual analogue scale, pressure algometry), disability (ODI, RDQ, FABQ)	Both interventions induced lower pain and self-perceived disability. Wii Fit induced significantly higher improvements for all outcomes except for deep tissue mechanical pain sensitivity (pressure algometry)
Adverse event: No	
Drug dependency problems	
Cutter et al. (2014), USA	Opioid- or cocaine-dependent subjects (F/M, N = 17/12)
Mean age: 43 ± 9 years	Intervention (RCT)
Two groups: Wii Fit vs. sedentary video games
Duration: eight weeks (20–25 min, five sessions/week)
Location: Drug rehabilitation center	For each session, subjects were invited to choose, two aerobics activities, one yoga activity, one balance activity and one strength activity	Acceptability (attendance, four-item questionnaire), physical activity (in-session energy expenditure, IPAQ-L), substance use (diary, urine toxicology screening), well-being (PSS, BLSS, LOT)	Both interventions showed high level of acceptability, decreased substance use and increased well-being. Wii Fit participants reported high level of MVPA at the end of the intervention period
Adverse events: No	
Cancer patients (fatigue management)	
Hoffman et al. (2013)2, USA	Post-surgical non-small lung cancer patients (F/M, N = 5/2)
Age range: 53–73 years
Mean age: 65 ± 7 years	Intervention (phase 1, cf. phase 2 below)
One group
Duration: six weeks (5–30 min “walking with the Wii” (see elsewhere) + 3–4 Wii Fit balance activities, five sessions/week)
Location: Home	Aerobics: “walking with the Wii” (might described an aerobics—Free Run—activity played by walking instead of running); balance: Ski Slalom, Soccer Heading; Training Plus: Driving Range; and “other activities”	Acceptability (questionnaire), fatigue (BFI), Fatigue management (PSEFSM), balance and functional mobility (ABC, self-efficacy for walking duration instrument, step-count) and adherence (playing time)	Patients adhered to the Wii Fit intervention, which was rated as acceptable. Perceived efficacy for balance and functional mobility increased, perceived fatigue decreased, and perceived self-efficacy for fatigue self-management increased (No statistics however). Light intensity home-based exertion delivered via a game console was effective for fatigue self-management in cancer patients
Adverse events: No	
Hoffman et al. (2014)2,USA	Same as for phase 1 (cf. above)	Intervention (phase 2, cf. phase 1 above)
One group
Duration: 10 weeks (30 min “walking with the Wii” (see elsewhere) + 3–4 Wii Fit balance activities, five sessions/week)
Location: Home
Note: phase 1 and phase 2 together: 16-week intervention	Same as for phase 1 (cf. above)	Same as for phase 1 (cf. above)	Positive outcomes noted at the end of phase 1 (cf. the above) were maintained or reinforced at the end of the phase 2. Light intensity home-based exertion delivered via a game console was effective for fatigue self-management in cancer patients (even for those undergoing an adjuvant therapy) for a period as long as 16 weeks at least.
Adverse events: No
ABC (no unit):
Wii Fit group
Pre-intervention: 72.8 ± 20.5
Post-intervention: 88.9 ± 24.8	
Systemic lupus erythematosus	
Yuen et al. (2011), USA	African American women with systemic lupus erythematosus (N = 15)
Age range: 25–67 years
Mean age: 47 ± 14 years	Intervention
One group
Duration: 10 weeks (30 min, two sessions/week)
Location: Home	Yoga, strength and aerobics activities	Primary: fatigue (FSS)
Secondary: anxiety level, pain intensity, body composition, step-count, physical fitness, adherence	Fatigue, anxiety and pain were reduced. Body composition and physical fitness improved. Good adherence
Adverse events: No	
Seniors	
Chan et al. (2012), China	Elderly referred for rehabilitation (F/M, N = 22/8, 13 of them having acquired neurological disorders)
Mean age: 80 ± 7 years	Intervention
One group (+comparison with a historic pool of 60 patients)
Duration: 5–9 weeks (10 min, 1–2 sessions/week, total of eight Wii Fit sessions)
Location: Geriatric center	Aerobics: 2P Run	Primary: feasibility (Borg scale, HR), adherence (playing time), acceptability (questionnaire)
Secondary: functional ability (FIM)	Participants completed an average of 72 ± 7 min of Wii Fit during their 5–9-week rehabilitation period (instructions: about 80 min). No difference was noted in exertion rate between the Wii Fit activity and a traditional arm ergometer exercise. Wii Fit participants exhibited a higher improvement in functional abilities compared to historic controls and wanted to continue the game at home
Adverse events: No	
Daniel (2012), USA	Pre-frail elderly (F/M, N = 14/9)
Mean age: 77 ± 5 years	Intervention (RCT)
Three groups: passive control vs. seated exercise training vs. Wii Fit + weight vest
Duration: 15 weeks (45 min, three sessions/weeks)
Location: Laboratory	Not specified (supposedly, all activities included in Wii Fit and Wii Sports)	Physical fitness (SFT, CHAMPS) body composition, balance and functional mobility (ABC, LLFDI, 8-feet TUG), adherence (attendance)	Authors described an improvement in physical fitness and balance confidence in the Wii Fit group, but no statistical significance was indicated. Same attendance rate (86%) in both seated exercising and Wii Fit groups.
Adverse events: No	
Notes:

%, Percentage; ABC, activities-specific balance confidence scale; BFI, brief fatigue inventory; BLSS, brief life satisfaction scale; BMI, body mass index; CESD, center for epidemiologic studies depression scale; CHAMPS, community healthy activities model program for seniors; COPD, chronic obstructive pulmonary disease; CRQ-SR, chronic respiratory questionnaire; CTSIB, clinical test of sensory interaction and balance; EE, energy expenditure; ESWT, endurance shuttle walk test; FABQ, fear avoidance beliefs questionnaire; FIM, functional independence measure; FFS, fatigue severity scale; HR, hear rate (beats/min); IMI, intrinsic motivation inventory; LLFDI, late life function and disability index; IPAQ-L, physical activity questionnaire-long version; LOT, life orientation test; METs, metabolic equivalent; MVPA, moderate-to-vigorous physical activity; PSEFSM, perceived self-efficacy for fatigue self-management; RPP, rate pressure product; PAID, problem areas in diabetes scale; sec, second; SEES, subjective exercise experience scale; SFT, senior fitness test; PA, physical activity; PACES, physical activity and exercise questionnaire; ODI, Oswestry low-back pain disability index; PSS, perceived stress scale; RDQ, Roland Morris disability questionnaire; SF-12, short form-12 health survey; SOT, sensory organization test; TD2M, type 2 diabetes mellitus; TUG, time up and go; VPA, vigorous physical activity; VO2, oxygen consumption; VO2max, maximal oxygen consumption; WHO-5, five-item WHO well-being index.

For the same test, unit may vary from one paper to another.

1 When balance outcomes were included concomitantly with other outcomes, and were not described as a primary outcome alone, the study was only included in Table 3.

2 Hoffman et al. (2013, 2014) report results from two different phases of the same project.

Table 4 Wii Fit interventions for functional balance training.

Authors (year), country	Population characteristics	Study design	Wii Fit activities (or other video games)	Outcomes and measures	Key findings and data used for the meta-analyses	
Healthy young adults or middle-aged adults	
Gioftsidou et al. (2013), Greece	Healthy young adults (F/M, N = 18/22)
Age range: 20–22 years
Mean age: 20 ± 1 years	Intervention (RCT)
Two groups: Wii Fit vs. BOSU ball-based therapy
Duration: eight weeks (14 min, two sessions/week)
Location: Laboratory (supposedly)	Balance: Balance Bubble, Penguin Slide, Snowboard Slalom, Ski Slalom, Soccer Heading, Table Tilt; Training plus: Balance Bubble Plus, Skateboard Arena, Table Tilt Plus	Balance (single leg stance tests1 and various indexes using Biodex system)	Balance improvements for both BOSU ball and Wii Fit intervention. Only one test (the balance board anterior-posterior single-limb stance test) showed greater improvement for the BOSU ball training
Adverse events: No	
Melong & Keats (2013), Canada	Healthy young adults (F/M, N = 12/8)
Mean age: 20 ± 1 years	Intervention (RCT)
Two groups: Wii Fit vs. BOSU ball-based therapy
Duration: four weeks (20 min, three sessions/week)
Location: Laboratory	Balance: Ski Jump, Ski Slalom, Soccer Heading, Table Tilt (+ a other activities played with the Wii Balance Board)	Primary: adherence (attendance and playing time)
Secondary: enjoyment (PACES), balance (stabilometry)	Balance improvement were noted in both groups. While the Wii Fit group showed higher levels of enjoyment, this did not lead to a significantly higher attendance or playing time. This study may have been underpowered
Adverse events: No	
Lee, Lee & Park (2014), Korea	Healthy young adults
(N = 24)
Mean age: 20 ± 1 years	Intervention (RCT)
Two groups: Wii Fit vs. indoor horseback riding exercise
Duration: six weeks (25 min, three sessions/week)
Location: not specified (« indoor »)	Balance: Balance Bubble, Ski Slalom, Table Tilt	Balance (dynamic tests using Biodex system: anteroposterior, mediolateral, and overall stability)	Both the Wii Fit and indoor horseback riding programs induce significant improvement in all three dynamic balance tests
Adverse events: No	
Cone, Levy & Goble (2015), USA	Healthy young adults
(F/M, N = 16/24)
Age range: 18–35 years
Mean age: 23 ± 3 years	Intervention (RCT)
Two groups: Wii Fit vs. passive control
Duration: six weeks (30–45 min, 2–4 sessions/week)
Location: Laboratory (supposedly)	Balance: Balance Bubble, Penguin Slide, Snowboard Slalom, Ski Slalom, Soccer Heading, Table Tilt, Tightrope Walk	Balance (SOT, LOF)	Significantly higher improvements in both LOS and SOT scores were noted for the Wii Fit group. Because those tests respectively focus on dynamic stability and sensory weighting, the results suggest that individuals with vestibular system alterations or dynamic balance control impairments may benefit from Wii Fit training
Adverse events: No	
Naumann et al. (2015), Germany	Healthy young adults (F/M, N = 29/8)
Age range: 20–34 years
Mean age: 23 ± 3 years	Intervention (RCT)
Three groups: Wii Fit vs. MFT Challenge Disc® vs. passive control
Duration: four weeks (30 min, three sessions/week) + follow-up after four weeks
Location: Laboratory (supposedly)	Balance: Balance Bubble, Ski Slalom, Snowboard Slalom, Table Tilt	Balance (game scores, single- or two-leg stance COP excursion)	The performance on trained games increased in both intervention groups. No changes were noted for the COP excursion tests. Similarly, the Wii Fit group did not show any increase in MFT Challenge Disc® scores, and vice-versa. These data suggest that the training effect of Wii Fit was highly specific and may not be transferred to real life balance-related tasks
Adverse events: No	
Healthy seniors	
Williams et al. (2011), USA	Elderly (F/M, N = 18/4)
Age range: 74–84 years
Mean age: 84 ± 5 years	Intervention
One group
Duration: four weeks (20 min, one session/week)
Location: Geriatric center-based	Balance and aerobics activities	Balance (BBS)	Wii Fit induced improvement in balance skills. The post-intervention BBS scores (49 ± 5) were significantly higher than pre-intervention scores (39 ± 6)
Adverse events: No
BBS (no unit):
Wii Fit group
Pre-intervention: 39.41 ± 6.28
Post-intervention: 48.55 ± 4.58	
Bateni (2012), USA	Elderly (F/M, N = 9/8)
Age range: 53–91 years
Mean age: 73 ± 14 years	Intervention (RCT)
Three groups: Wii Fit vs. Wii Fit + traditional physical therapy vs. traditional physical therapy alone
Duration: four weeks (15 min—estimation, three sessions/week)
Location: Rehabilitation center	Balance: Balance Bubble, Ski Jump, Ski Slalom; training plus: Table Tilt plus	Balance (BBS and Wii Fit Balance Bubble score)	Improvements in both BBS and Balance Bubble score in all three groups were observed. However, subjects who underwent traditional therapy exercises performed better at the BBS compared to subjects who only play Wii Fit alone
Adverse events: No	
Franco et al. (2012), USA	Elderly (F/M, N = 25/7)
Mean age: 78 ± 6 years	Intervention (RCT)
Three groups: Wii Fit + strength training vs. traditional balance training vs. passive control
Duration: three weeks (10–15 min, two sessions/week)
Location: Community dwelling	Balance: Ski Jump, Ski Slalom, Soccer Heading, Table Tilt, Tightrope Walk	Balance and gait (BBS, Tinetti test), functional health and well-being (SF-36), enjoyment (home-made questionnaire) and adherence (playing time)	Wii Fit did not induce any balance and gait improvements. Same outcomes were observed in the traditional training group. Subjects playing Wii Fit reported high level of enjoyment.
Adverse events: No
BBS (no unit):
Wii Fit group
Pre- and post-intervention delta: 3.55 ± 5.03
Traditional therapy group
Pre- and post-intervention delta: 3.45 ± 2.50	
Orsega-Smith et al. (2012), USA	Elderly (F/M, N = 30/4)
Age range: 55–86 years
Mean age: 72 ± 8 years	Intervention (No-RCT)
Three groups: playing Wii Fit for four weeks vs. playing Wii Fit for eight weeks vs. passive controls
Duration: four or eight weeks (30 min, two sessions/week)
Location: Community dwelling	Yoga: Deep Breathing, Half Moon, Palm Tree; aerobics: Hula Hoop; balance: Balance Bubble, Penguin Slide, Snowboard Slalom, Ski Jumping, Ski Slalom, Table Tilt	Balance (BBS), mobility (8-feet TUG), leg strength (STST2), balance confidence (ABC, FES), autonomy (ADL)	Balance, and ability to complete activities of daily living were improved in the two Wii Fit groups. Leg strength increased in the four-week intervention group only, while balance confidence increased in the eight-week intervention group only. No change was noted in the control group
Adverse events: No
ABC (no unit):
Wii Fit group (A: four-week)
Pre- and post-intervention delta: 3.20 ± 13.88
Wii Fit group (B: eight-week)
Pre- and post-intervention delta: 6.07 ± 7.04
BBS (no unit):
Wii Fit group (A: four-week)
Pre- and post-intervention delta: 1.44 ± 2.34
Wii Fit group (B: eight-week)
Pre- and post-intervention delta: 1.22 ± 1.09	
Rendon et al. (2012), USA	Elderly (F/M, N = 26/14; six using an assistive devise)
Age range: 60–95 years
Mean age: 85 ± 5 years	Intervention (RCT)
Two groups: Wii Fit vs. passive control
Duration: six weeks (35–45 min, three sessions/week)
Location: Community dwelling	Strength: Lunge, Single Leg Extension, Single Leg Twist	Primary: balance (ABC, 8-feet up and go test)
Secondary: depression (GDS)	Wii Fit improved dynamic balance and balance confidence. No effect on depression score
Adverse events: No	
Toulotte, Toursel & Olivier (2012), France	Elderly (some had an history of falling; F/M, N = 22/14)
Age range: >60 years
Mean age: 75 ± 10 years	Intervention (RCT)
Four groups: passive control vs. Wii-Fit, adapted physical activities vs. Wii Fit + adapted physical activities
Duration: 20 weeks (60 min, one session/week)
Location: Fitness room	Yoga activities and some balance activities (Soccer Heading, Ski Jump, Ski Slalom, Tightrope Walk, and another activity identified as “game balls”)	Balance (static tests only: a single leg stance test1 and the Wii Fit balance test; static and dynamic test: Tinetti test)	Wii Fit significantly improved static balance but not dynamic balance. The conventional adapted PA training improved both. Combining both interventions did not induce additional benefits
Adverse events: No	
Bieryla & Dold (2013), USA	Elderly (F/M, N = 10/2)
Age range: 70–92 years
Mean age: 82 ± 6 years	Intervention (RCT)
Two groups: Wii Fit vs. passive control
Duration: three weeks (30 min, three sessions/week) + follow-up at four-week
Location: Community dwelling (supposedly: “supervised”)	Yoga: Chair, Half Moon, Warrior; aerobics: Torso Twists; balance: Ski Jump Soccer, Heading	Balance (BBS, TUG, FAB, functional reach test) and adherence (retention rate)	In the Wii Fit group, the retention rate was 4/6 at the four-week follow-up. The Wii Fit training induced an improvement in the BBS only
Adverse events: No	
Chao et al. (2013), USA	Assisted living residents (F/M, N = 5/2, three of them having acquired neurological disorders)
Age range: 80–94 years
Mean age: 86 ± 5 years	Intervention
One group
Duration: eight weeks (30 min, two sessions/week)
Location: Assisted living dwelling	Yoga: Chair, Deep Breathing; strength: Lunge; aerobics: Basic Run; balance: Penguin Slide, Table Tilt	Balance and mobility (BBS, TUG, 6 min walk test, FES), perceived efficacy (SSE, OEE), acceptability (questionnaire), safety	The Wii Fit intervention was acceptable and safe, and induced significant improvements in BBS. Trends only (p = 0.06) were noted for improvement in other balance and mobility indexes
Adverse events: No
BBS (no unit):
Wii Fit group
Pre-intervention: 40.9 ± 8.5
Post-intervention: 45.1 ± 8.3
TUG (sec):
Wii Fit group
Pre-intervention: 19.4 ± 5.5
Post-intervention: 15.8 ± 5.1	
Janssen, Tange & Arends (2013), The Netherlands	Home nursing residents (F/M, N = 20/9, some had a history of falling)
Average age: 65–90 years
Mean age: 82 ± 9 years	Intervention (No-RCT)
Three groups: Wii Fit (without history of playing) vs. Wii Fit (with an history of playing) vs. passive control
Duration: 12 weeks (10–15 min, two sessions/week)
Location: Assisted living dwelling	Table Tilt Plus (Training Plus) and two other games of participants’ choice	Primary: balance (BBS)
Secondary: physical activity (LASAPAQ)	No significant balance improvements in either Wii Fit intervention groups. However, subjects increased their volume of physical activity by about 60 min/day
Adverse events: No	
Chao et al. (2014), USA	Assisted living resident
(F/M, N = 24/8)
Mean age: 85 ± 6 years	Intervention (RCT)
Two groups: Wii Fit vs. “education” semi-passive control
Duration: four weeks (30 min, two sessions/week)
Location: Assisted living dwelling	Yoga: Chair, Deep Breathing; strength: Lunge; balance: Penguin Slide, Table Tilt; aerobics: Basic Run	Balance and physical function (BBS, TUG, 6 min walk test, FES, SEE), depression (GDS), quality of life (SF-8)	Significant improvements in balance related-function and depression parameters were found in the Wii Fit group only. Wii Fit might be considered as a potential activity for older adults in assisted living dwellings
Adverse events: No
BBS (no unit):
Wii Fit group
Pre-intervention: 40.53 ± 6.59
Post-intervention: 43.93 ± 6.34
TUG (sec):
Wii Fit group
Pre-intervention: 18.52 ± 5.60
Post-intervention: 15.27 ± 4.68	
Cho, Hwangbo & Shin (2014), Korea	Elderly (N = 32)
Mean age: 78 ± 1 years	Intervention (RCT)
Two groups: Wii Fit vs. passive control
Duration: eight weeks (30 min, three sessions/week)
Location: not specified	Balance: Balance Bubble, Ski Slalom, Table Tilt	Balance (Romberg test)	Significant improvements were noted in the Wii Fit group only
Adverse events: No	
Nicholson et al. (2015), Australia	Elderly (F/M, N = 14/27)
Age range: 65–84 years
Mean age: 75 ± 5 years	Intervention (RCT)
Two groups: Wii Fit vs. usual exercise
Duration: six weeks (30 min, three sessions/week)
Location: Community dwelling	Balance: Penguin Slide, Ski Jump, Ski Slalom, Soccer Heading, Table Tilt, Tightrope Walk; training plus: Snowball Fight, Perfect 10	Balance and mobility (TUG, functional reach tests, single leg stance test,1 STST,2 Icon-typed FES, walking speed), enjoyment (PACES), adherence (playing frequency)	The Wii Fit group showed more improvement compared to the control group for the followings: TUG, lateral reach, left-leg single leg stance test and gait speed. Interestingly, enjoyment increased during the intervention and the adherence was very high (average attendance: 17.5 out of 18 recommended sessions)
Adverse events: exacerbations of lower back pain (N = 2)
TUG (sec):
Wii Fit group
Pre- and post-intervention delta: −0.61 ± 0.79
Traditional therapy group
Pre- and post-intervention delta: −0.14 ± 0.88	
Roopchand-Martin et al. (2015), Jamaica	Elderly (F/M, N = 26/7)
Mean age: 70 ± 7 years	Intervention
One group
Duration: six weeks (30 min, two sessions/week)
Location: Community dwelling	Yoga: Tree; balance: Balance Bubble, Penguin Slide, Snowboard Slalom, Soccer Heading, Table Tilt; training plus: Obstacle Course, Skateboard	Balance and function (BBS, MCTSIB, MDRT, SEBT)	Significant balance and functional improvements were noted at the end of the Wii Fit intervention (i.e. for BBS, MDRT, SEBT, but not for MCTSIB)
Adverse events: No
BBS (no unit):
Wii Fit group
Pre- and post-intervention delta: 1.54 ± 2.60	
Seniors presenting balance impairments	
Williams et al. (2010), UK	Elderly with an history of falling (N = 21)
Mean age: 77 ± 5 years	Intervention (RCT)
Two groups: Wii Fit vs. standard care
Duration: 12 weeks (two sessions/week)
Location: Rehabilitation center	Yoga: Deep Breathing; aerobics: Basic Step, Hula Hoop, Running activities; Balance: Ski Jump, Ski Slalom, Soccer Heading, Table Tilt,	Functional balance (BBS, Tinetti test), static balance (Wii Fit age test) Balance confidence (FES), acceptability (retention, playing frequency, interviews)	The intervention met a high rate of acceptability. Balance improved in the Wii Fit group only
Adverse events: fall (N = 1, no injury)
BBS (no unit):
Wii Fit group
Pre-intervention: 43.7 ± 9.5
Post-intervention: 44.8 ± 11.8
Traditional therapy group
Pre-intervention: 36.3 ± 9.9
Post-intervention: 39.0 ± 10.2	
Agmon et al. (2011), USA	Elderly with balance impairment (F/M, N = 4/3)
Age range: 78–92 years
Mean age: 84 ± 5 years	Intervention
One group
Duration: 13 weeks (30 min, three sessions/week)
Location: Home	Balance: Basic Step, Ski Slalom, Soccer Heading, Table Tilt	Primary: Balance (BBS)
Secondary: mobility (4 m walk test), enjoyment (PACES), feasibility (playing time), safety (interviews)	The Wii Fit intervention increased balance and mobility. Some activities were more enjoyable than the others. Adherence was associated with enjoyment.
Adverse events: hip and neck strain (N = 1 and 2, respectively)
BBS (no unit):
Wii Fit group
Pre-intervention: 49.0 ± 2.1
Post-intervention: 53.0 ± 1.8	
Bainbridge et al. (2011), USA	Elderly with perceived balance deficit (F/M, N = 7/1)
Age range: 65–87 years
Mean age: 75 ± 8 years	Intervention
One group
Duration: six weeks (30 min, two sessions/week)
Location: Community dwelling	Yoga: Half Moon, Warrior; Balance: Penguin Slide, Ski Jump, Ski Slalom, Soccer Heading, Table Title, Tightrope Walk	Balance (BBS, ABC, MDRT), COP excursion measurements and other parameters (ankle range of motion tests…)	No statistically significant changes, but four patients (over the six who finished the intervention) demonstrated improvements on the BBS, based on established clinical guidelines
Adverse events: No	
Jorgensen et al. (2013), Denmark	Elderly with perceived balance deficit (F/M, N = 40/18)
Mean age: 75 ± 6 years	Intervention (RCT)
Two groups: Wii Fit vs. passive control
Duration: 10 weeks (35 min, two sessions/week)
Location: Community dwelling	Balance: Penguin Slide, Ski Slalom, Table Tilt, Tightrope Walk; training plus: Perfect 10	Primary: strength (maximal voluntary contraction of leg extensors (MVC), Rate of Force Development (RFD), static balance (COP velocity moment)
Secondary: mobility (TUG, STST2), balance confidence (FES), motivation (questionnaire)	Compared to controls, the Wii Fit group exhibited increased strength after 10 weeks of training. Mobility and balance confidence parameters also showed an improvement in the Wii Fit group only. Motivation for Wii Fit training was found to be high.
Adverse events: No
TUG (sec):
Wii Fit group
Pre-intervention: 10.3 ± 3.8
Post-intervention: 9.0 ± 3.2	
Seniors with acquired neurological alterations	
dos Santos Mendes et al. (2012), Brazil	Patients with Parkinson’s disease (stages 1 and 2 on Hoehn & Yahr scale, N = 16)
Mean age: 69 ± 8 years	Intervention
Two groups: patients vs. healthy controls, N = 11)
Duration: seven weeks (20–30 min, two sessions/week) + follow-up after two months
Location: Rehabilitation center (supposedly: « Overseen by a physiotherapist »)	Strength: Single Leg Extension, Torso Twists; aerobics: Basic Step; balance: Penguin Slide, Soccer Heading, Table Tilt; training plus: Basic Run Plus, Obstacle Course, Rhythm Parade, Tilt City	Stability (functional reach test) and motor learning (score performed in the selected games before and after the intervention)	Seven of the 10 tested games induced the same learning in Parkinson’s disease patients compared with healthy subjects. These patients were also able to transfer and retained (+2-months follow-up) their learning on a similar but untrained functional task
Adverse events: No	
Esculier et al. (2012)3, Canada	Patients with Parkinson’s disease (F/M, N = 5/6)
Age range: 48–80 years
Mean age: 62 ± 11 years	Intervention
Two groups: patients vs. healthy controls, F/M, N = 4/5)
Duration: six weeks (40 min, three sessions/week)
Location: Home	Yoga: Deep Breathing; aerobics: Hula-Hoop; balance: Balance Bubble, Penguin Slide, Ski Jump, Ski Slalom, Table Tilt	Functional balance and mobility (ABC, STST,2 TUG, Tinetti test, 10 m walk test, CBM), static balance (single leg stance test,1 COP excursion)	Improvements in every outcome (except for ABC) in the two groups. A home-based Wii Fit improved static and dynamic balance, mobility and functional abilities of people affected by Parkinson’s disease
Adverse events: No	
Pompeu et al. (2012), Brazil	Patients with Parkinson’s disease (stages 1 and 2 on Hoehn & Yahr scale, F/M, N = 15/17)
Age range: 60–85 years
Mean age: 67 ± 8 years	Intervention (RCT)
Two groups: Wii Fit vs. traditional balance training
Duration: seven weeks (30 min, two sessions/week) + follow-up 60 days after
Location: Rehabilitation center	Strength: Single Leg Extension, Torso Twist; aerobics: Basic Step, Basic Run; balance: Penguin Slide, Soccer Heading, Table Tilt; training plus: Obstacle Course, Rhythm Parade, Tilt City	Primary: performance in daily activities
Secondary: balance (static: single leg stance test,1 dynamic: BBS), cognition (Montreal cognitive assessment)	Same improvements in Wii Fit and traditional balance training groups (maintained at 60 days follow-up). No additional advantage for the Wii Fit group
Adverse events: No
BBS (no unit):
Wii Fit group
Pre- and post-intervention delta: 1.4 ± 2.6
Traditional therapy group
Pre- and post-intervention delta: 1.1 ± 2.1	
Padala et al. (2012), USA	Patients with an history of mild Alzheimer’s Dementia (F/M, N = 16/6)
Mean age: 80 ± 7 years	Intervention (RCT)
Two groups: Wii Fit vs. walking
Duration: eight weeks (30 min, five sessions/week)
Location: Assisted living center	Yoga: Chair, Half moon, Sun Salutation
Warrior; strength: Lunge, Single Leg Extension, Torso
Twist; balance: Balance Bubble, Penguin Slide, Ski Jump, Ski Slalom,
Soccer Heading,
Table Tilt	Primary: balance (BBS, TUG, Tinetti test)
Secondary: functional ability (ADL and instrumental ADL), quality of life (quality of life in Alzheimer’s disease scale), cognition (mini mental state examination)	Significant improvements for balance outcomes in the Wii Fit group only (trends for the walking group). No significant changes in other outcomes, except for quality of life (walking group only)
Adverse events: No
BBS (no unit):
Wii Fit group
Pre-intervention: 43.4 ± 8.9
Post-intervention: 47.5 ± 5.9
Traditional therapy group
Pre-intervention: 41.3 ± 7.6
Post-intervention: 46.9 ± 6.3
TUG (sec):
Wii Fit group
Pre-intervention: 14.7 ± 7.2
Post-intervention: 14.3 ± 6.8
Traditional therapy group
Pre-intervention: 14.9 ± 4.7
Post-intervention: 12.8 ± 3.2	
Barcala et al. (2013), Brazil	Hemiplegic stroke patients (F/M, N = 11/9)
Age range:
Mean age: 64 ± 14 years	Intervention (RCT)
Two groups: conventional therapy + Wii Fit vs. conventional therapy + balance training
Duration: five weeks (30 min, two sessions/week)
Location: Rehabilitation center	Balance: Penguin Slide, Table Tilt, Tightrope Walk	Functional balance (BBS), static balance (stabilometry), functional mobility, independence (TUG, functional independence test)	Both groups showed significant improvements in all parameters. No statistical differences were noted between the two groups emphasizing the efficacy of the Wii Fit therapy for functional recovery in hemiplegic stroke patient
Adverse events: No
BBS (no unit):
Wii Fit group
Pre-intervention: 39.6 ± 6.43
Post-intervention: 41.9 ± 6.91
Traditional therapy group
Pre-intervention: 37.2 ± 5.22
Post-intervention: 42.2 ± 4.8
TUG (sec):
Wii Fit group
Pre-intervention: 27.9 ± 8.22
Post-intervention: 24.3 ± 8.64
Traditional therapy group
Pre-intervention: 28.1 ± 3.10
Post-intervention: 25.2 ± 2.78	
Mhatre et al. (2013), USA	Patients with Parkinson’s disease (stages 2.5 or 3 on Hoehn & Yahr scale, F/M, N = 6/4)
Age range: 44–91 years
Mean age: 67 years	Intervention
One group
Duration: eight weeks (30 min, three sessions/week)
Location: Rehabilitation center	Balance: “marble tracking,” “skiing,” “bubble rafting” (possibly: Table Tilt, Ski Slalom and Balance Bubble)	Primary: Balance (BBS; DGI; Sharpened Romberg; Wii Balance Board-assisted postural sway tests)
Secondary: Balance (ABC) and depression (GDS)	Significant improvements in BBS (3.3) and some other balance & gait outcomes, but not in balance confidence (ABC) or mood (GDS)
Adverse events: No
ABC (no unit):
Wii Fit group
Pre-intervention: 83.5 ± 5.3
Post-intervention: 82.5 ± 3.6
BBS (no unit):
Wii Fit group
Pre-intervention: 48.8 ± 3.2
Post-intervention: 52.1 ± 2.3	
Bower et al. (2014), Australia	Stroke inpatients (F/M, N = 13/17)
Mean age: 64 ± 15 years	Intervention (RCT)
Two groups: Wii Fit balance training vs. Wii Sports upper limb training
Duration: 2–4 weeks (45 min, three sessions/week)
Location: Rehabilitation center	A selection of 18 activities among the 66 activities proposed in the Wii Fit Plus software (including Deep Breathing, Ski Slalom, Basic Run and others…)	Primary: adherence (retention, attendance, playing time), acceptability (recruitment rate, questionnaire), safety (questionnaire)
Secondary: Balance (Step Test, Wii Balance Board Test), Functional autonomy (functional reach test, upper limb—motor assessment scale), mobility (TUG, STREAM), balance confidence (FES)	The recruitment rate (21%), eligibility rate (86%), retention rate (90% and 70%, respectively, at two and four weeks) and adherence rate (99% and 87%) indicated that a Wii Fit intervention would be feasible in stroke inpatients. All the patients enjoyed the intervention, which was described as safe. However, trends only were noted for improvements in some of the balance tests.
Adverse events: Falls (N = 4), no subsequent injury
TUG (sec):
Wii Fit group
Pre- and post-intervention delta: −11.2 ± 10.3	
Esculier, Vaudrin & Tremblay (2014)3, Canada	Patients with Parkinson’s disease (stages 3.5 or more on Hoehn & Yahr scale, F/M, N = 3/5)
Mean age: 64 ± 12 years	Intervention
Two groups: patients vs. healthy controls (F/M, N = 3/5)
Duration: six weeks (40 min, three sessions/week)
Location: Home	A selection of balance and strength activities involving lower limb muscles (i.e. using semi-squats Positions)	Lower limb corticomotor excitability (transcranial magnetic stimulation)	Wii Fit training improved lower limb corticomotor excitability in Parkinson’s patients. Depending on the experimental conditions, these improvements were similar or more important when compared to healthy subjects. Home-based interventions including visual feedbacks could be beneficial for functional improvement in Parkinson’s patients
Adverse events: No	
Goncalves et al. (2014), Brazil	Patients with Parkinson’s disease (stages 2–4 on Hoehn & Yahr scale, F/M, N = 8/7)
Mean age: 69 ± 10 years	Intervention
One group
Duration: seven weeks (40 min, two sessions/week)
Location: Hospital (supposedly, not specified)	Balance: Ski Jump, Ski Slalom, Soccer Header;
aerobics: Free Step, Rhythm Boxing;
training plus: Island Cycling, Rhythm Parade (supposedly), Segway Circuit
(+2 other activities that were not explicitly named)	Functional mobility (UPDRS, SE, FIM), gait (number of steps, walking speed)	The Wii Fit program induced gait improvement, but statistical significance was not indicated. Functional mobility was significantly improved (i.e., decrease in UPDRS score, and increase in SE and FIM scores)
Adverse events: No	
Hung et al. (2014), China	Chronic stroke patients
(F/M, N = 10/18)
Mean age: 54 ± 10 years	Intervention (RCT)
Two groups: Wii Fit vs. traditional weight-shift training
Duration: 12 weeks (30 min, Two sessions/week) + follow-up at three months
Location: rehabilitation center (supposedly: according to pictures, “supervised by an occupational therapist”)	Yoga: Warrior; balance: Balance Bubble, Penguin Slide, Ski Slalom, Soccer Heading, Table Tilt; aerobics: Basic Step	Balance (a series of COP excursion tests, FES), function (forward reach, TUG), enjoyment (PACES)	At the end of the intervention, Wii Fit induced a higher increase in some COP excursion tests compared to the traditional weigh-shift training group. However, at a three-month follow-up, these effects were not maintained, while the traditional weight-shift group showed higher improvements. Both types of intervention showed significant improvements in balance and functional outcomes, and the enjoyment was higher in the Wii Fit group
Adverse events: No
TUG (sec):
Wii Fit group
Pre-intervention: 26.06 ± 12.05
Post-intervention: 20.88 ± 7.77
Traditional therapy group
Pre-intervention: 29.45 ± 16.22
Post-intervention: 26.61 ± 12.92	
Liao et al. (2015), China	Patients with Parkinson’s disease
(stages 1 to 2 on Hoehn & Yahr scale; F/M, N = 19/17)
Mean age: 66 ± 7 years	Intervention (RCT)
Three groups: Wii Fit vs. traditional therapy vs. passive control
Duration: six weeks (45 min, two sessions/week + follow-up after one month
Location: Rehabilitation center (supposedly: “administrated by the same physical therapist”)	Yoga (10 min): Chair, Sun Salutation, Tree; strength (15 min); balance (20 min): Balance Bubble, Soccer Heading, Ski Slalom, Table Tilt	Primary: mobility (obstacle crossing performance tests measured with the Liberty system), dynamic balance (LOS)
Secondary: balance (SOT, FES), mobility (TUG), quality of life (PDQ-39)	When compared with the passive control group, Wii Fit induced significant increases for the mobility, balance and quality of life outcomes. Interestingly, movement velocity evaluated with LOS test showed significantly greater improvement in the Wii Fit group compared to traditional therapy. These results should encourage the implementation of Wii Fit activities in patients with Parkinson’s disease.
Adverse events: No
TUG (sec):
Wii Fit group
Pre- and post-intervention delta: −2.9 ± 2.2
Traditional therapy group
Pre- and post-intervention delta: −1.1 ± 0.1
Passive control group
Pre- and post-intervention delta: +0.7 ± 1.7	
Morone et al. (2014), Italy	Subacute stroke patients (N = 50)
Mean age: 60 ± 10 years	Intervention (RCT)
Two groups: Wii Fit + traditional therapy vs. traditional balance exercises + traditional therapy
Duration: four weeks (20 min, three sessions/week) + follow-up after one month
Location: Rehabilitation center	Balance: Balance Bubble, Ski Slalom; aerobics: Hula Hoop	Primary: balance (BBS)
Secondary: mobility (10 m walk test, functional ambulatory category), independency (Barthel index)	Wii Fit was more effective than traditional balance exercises to improve balance and independency in subacute stroke patients. No significant differences were noted between groups for mobility outcomes (increase in both groups). Interestingly, benefits in balance ability were maintained one month after the intervention
Adverse events: No	
Omiyale, Crowell & Madhavan (2015), USA	Hemiparetic stroke patients (F/M, N = 4/6)
Age range: 41–73 years
Mean age: 67 ± 8 years	Intervention
One group
Duration: three weeks (60 min, three sessions/week)
Location: not specified, but “supervised by a physical therapist”	Balance: Balance Bubble, Penguin Slide, Ski Slalom, Table Tilt, Tightrope Walk	Neural plasticity (interhemispheric symmetry through tibialis anterior corticomotor excitability using transcranial magnetic stimulation), balance, motor response and function (COP distribution and dynamic weight shifting, Soccer Heading’s score, BBS, TUG, and dual TUG, gait speed, ABC)	Interestingly, the Wii Fit intervention significantly improved the interhemispheric symmetry. Overall, but not for all parameters, patients also improved their balance abilities, motor responsiveness, and balance related functions. These results suggest that Wii Fit rehabilitation may be able to influence positively neural plasticity and functional recovery in chronic stroke patients
Adverse events: No
ABC (no unit):
Wii Fit group
Pre-intervention: 65.9 ± 13.49
Post-intervention: 73.4 ± 13.32
BBS (no unit):
Wii Fit group
Pre-intervention: 51.6 ± 5.97
Post-intervention: 53.6 ± 2.95
TUG (sec):
Wii Fit group
Pre-intervention: 21.0 ± 12.18
Post-intervention: 19.4 ± 9.10	
Yatar & Yildirim (2015), Cyprus/Turkey	Chronic stroke patients (F/M, N = 13/17)
Mean age: 60 ± 14 years	Intervention (RCT)
Two groups: neurodevelopmental training + Wii Fit vs. neurodevelopmental training + progressive balance training
Duration: four weeks (30 min, three sessions/week)
Location: Rehabilitation center (supposedly)	Balance: Balance Bubble, Ski Slalom, Soccer Heading	Primary: static balance (Wii Balance Board-assisted postural sway tests), dynamic balance (BBS, DGI, functional reach test, TUG)
Secondary: balance confidence (ABC, ADL)	Primary and secondary outcomes increased in both Wii Fit and progressive balance training groups. The increment was statistically higher in the Wii Fit group for: DGI. Functional reach test and ABC. Large differences in baseline values between the two groups limits the interpretation
Adverse events: No
BBS (no unit):
Wii Fit group
Pre-intervention: 45.60 ± 5.26
Post-intervention: 50.33 ± 4.09
Traditional therapy group
Pre-intervention: 39.60 ± 9.31
Post-intervention: 44.80 ± 7.48
TUG (sec):
Wii Fit group
Pre-intervention: 17.96 ± 7.77
Post-intervention: 16.17 ± 8.23
Traditional therapy group
Pre-intervention: 26.36 ± 11.60
Post-intervention: 22.11 ± 11.88	
Orthopedic population	
Fung et al. (2012), Canada	Adult outpatients following knee replacement (F/M, N = 33/17)
Mean age: 68 ± 11 years	Intervention (RCT)
Two groups: traditional therapy + Wii Fit vs. traditional therapy + additional lower limb exercise
Duration: until discharge (≈six weeks, 15 min/session)
Location: Rehabilitation center	Yoga: Deep Breathing, Half Moon; Strength: Torso Twist; Aerobics: Hula Hoop; Balance: Balance Bubble, Penguin Slide, Ski Slalom, Table tilt, Tightrope Walk	Function (range of motion), 2 min walk test, LEFS), pain (NPRS), Balance confidence (ABC) and length of rehabilitation	From baseline to discharge, the improvements were similar between the two groups for all the outcomes. The Wii Fit intervention might induce higher improvement for the LEFS. But the study was not powerful enough to obtain significance
Adverse events: No	
Baltaci et al. (2013), Turkey	Young adults with anterior cruciate ligament reconstruction (N = 30)
Mean age: 29 ± 5 years	Intervention (RCT)
Two groups: Wii vs. conventional rehabilitation
Duration: 12 weeks (60 min, three sessions/week)
Location: Rehabilitation center (supposedly)	Not clear. Probably a combination of Wii Sports games (Bowling, Boxing) and Wii Fit activities (“skiing games,” “football,” “balance board”)	Balance (SEBT), function (functional squat test including coordination, proprioception, time response and strength measurements)	No difference between Wii Fit and conventional physical therapy. Wii Fit may be able to address rehabilitation goals for patients with anterior cruciate ligament reconstruction
Adverse events: No	
Sims et al. (2013), USA	Young active adults with an history of lower limb injury within one year (F/M, N = 16/12)
Mean age: 22 ± 2 years	Intervention (RCT)
Three groups: Wii Fit, traditional balance training, passive control
Duration: four weeks (15 min, three sessions/week)
Location: Rehabilitation center	Yoga: Chair, Half Moon, Tree; Balance: Balance bubble, Penguin Slide, Ski Slalom, Soccer Heading, Table Tilt; Strength: Lunge, Sideways Leg Lift, Single Leg Extension; Aerobics: Basic Step, Hula Hoop, Super Hula Hoop	Primary: balance (static: Time to Boundary test, dynamic: SEBT)
Secondary: function (LEFS)	Wii Fit improved static balance to a larger extend than the traditional balance training. Dynamic balance was improved in all groups
Adverse events: No	
Punt et al. (2015), Switzerland	Adults ankle sprain patients (F/M, N = 39/51)
Mean age: 34 ± 11 years	Intervention (RCT)
Three groups: Wii Fit vs. traditional balance training vs. passive control
Duration: six weeks (30 min, two sessions/week)
Location: Home	Balance: Balance Bubble, Penguin Slide, Ski Slalom, Table Tilt	Function (FAAM), pain (visual analogue scale), time to return to sport, satisfaction (questionnaire)	Foot and ankle ability score increased and pain decreased in all groups. A Wii Fit intervention was as effective as traditional therapy or no therapy. In the Wii Fit group, the average time to return to sport was 27 ± 20 days and 82% of patients were satisfied
Adverse events: No	
Multiple sclerosis	
Brichetto et al. (2013), Italia	Patients with multiple sclerosis (F/M, N = 22/14)
Mean age: 42 ± 11 years	Intervention (RCT)
Two groups: Wii Fit vs. traditional rehabilitation
Duration: four weeks (60 min, three sessions/week)
Location: Rehabilitation center	Balance: Lotus Focus, Ski Slalom, Snowboard Slalom, Soccer Heading, Table Tilt, Tightrope Walk	Primary: balance (BBS)
Secondary: fatigue (MFIS), posture (stabilometry)	More important balance improvements in the Wii Fit group. Fatigue was reduced and posture improved. A Wii Fit-based program might be more efficient than the standard rehabilitation procedure in multiple sclerosis patients
Adverse events: No
BBS (no unit):
Wii Fit group
Pre-intervention: 49.6 ± 4.9
Post-intervention: 54.6 ± 2.2
Traditional therapy group
Pre-intervention: 48.7 ± 3.3
Post-intervention: 49.7 ± 3.9	
Nilsagård, Forsberg & von Koch (2013), Sweden	Patients with multiple sclerosis (F/M, N = 64/20)
Mean age: 50 ± 11 years	Intervention (RCT)
Two groups: Wii Fit vs. passive control
Duration: 6–7 weeks (30 min, two sessions/week)
Location: Rehabilitation Center (“physical therapist supervised session”)	Aerobics: Skateboard Arena
Balance: Balance Bubble, Penguin Slide, Ski Slalom, Soccer Heading, Snowboard Slalom, Table Tilt, Tightrope Walk; Training plus: Balance Bubble plus, Perfect 10, Table Tilt Plus	Primary: balance (TUG)
Secondary: other functional tests (TUGcognitive, four square step test, 25-foot walk test, DGI, MSWS-12, ABC, STST2)	Improvement in several balance-related outcomes for the Wii Fit group. However, same improvements were observed in the controls (because of spontaneous exercise). Wii Fit can be recommended in adults with multiple sclerosis
Adverse events: No
ABC (no unit):
Wii Fit group
Pre- and post-intervention delta: 5.0 ± 14.4
TUG (sec):
Wii Fit group
Pre- and post-intervention delta: −0.8 ± 2.4	
Prosperini et al. (2013), Italy	Patients with multiple sclerosis (F/M, N = 25/11)
Mean age: 36 ± 9 years	Intervention (cross-over RCT)
Two groups: Wii Fit vs. passive control
Duration: 12 weeks (30 min, five sessions/week)
Location: Home	Supposedly all balance activities included in Wii Fit Plus (for the 4 first weeks, patients were allowed to play Zazen, Table Tilt and Ski Slalom only)	Balance (COP excursion, four-step square test), mobility (25-foot walk test), self-perceived disability (MSIS-29)	Wii Fit was effective in improving balance, mobility and self-perceived health status and quality of life
Adverse events: knee pain (N = 2), lower back pain (N = 3)	
Robinson et al. (2015), UK	Patients with multiple sclerosis (F/M, N = 38/18)
Mean age: 52 ± 6 years	Intervention (RCT)
Three groups: Wii Fit vs. traditional therapy vs. passive control
Duration: four weeks (40–60 min, two sessions/week)
Location: Rehabilitation center	Balance: Heading Soccer, Ski Slalom, Table Tilt, Tightrope Walk; Strength: Rowing Squats, Torso Twist; aerobics: Boxing, Hula Hoop, Advanced Steps (supposedly)	Primary: balance (postural sway), gait (gait speed), acceptability (UTAUT, FSS)
Secondary: self-perceived disability (MSWS-12, WHODAS)	Balance but not gait was improved by both the Wii Fit and traditional therapy interventions. Wii Fit was acceptable and induced positive changes in self-perceived disability
Adverse events: No	
Spinal cord injury patients	
Wall et al. (2015), USA	Individuals with incomplete spinal cord injury (>1-year post-injury, M, N = 5)
Age range: 50–64 years
Mean age: 59 ± 5 years	Intervention
One group
Duration: seven week (60 min, two sessions/week) + follow up after four weeks
Location: Home or University (and “supervised”)	Balance: Balance Bubble, Basic Run (or another running activity), Penguin Slide, Ski Slalom, Ski Jump, Tightrope Walk, Table Tilt; training plus: Island Bike, Obstacle Course, Segway Circuit	Walking ability (gait speed), balance (TUG, functional reach), well-being (SF-36)	Gait speed and functional reach tests score both significantly increased after the Wii Fit intervention and the effects were maintained at a four-week follow-up. However, the program failed to induce statistical improvements in wellness (SF-36) and TUG
Adverse events: No	
Children with developmental delay	
Salem et al. (2012), USA	Children with developmental delay (F/M, N = 18/22)
Age range: 39–58 months
Mean age: 49 ± 6 months	Intervention (RCT)
Two groups: Wii vs. passive control
Duration: 10 weeks (30 min, two sessions/week)
Location: Rehabilitation center	Strength: Lunges, Single Leg Stance; aerobics: Basic Run, Basic Step, Hula Hoop; balance: Penguin Slide, Soccer Heading, Tightrope Walk; Wii Sports’ Baseball, Bowling and Boxing games were also used	Primary: balance and gait (gait speed, TUG, single leg stance test,1 STST,2 TUDS, 2 min walk test)
Secondary: grip strength (dynamometer)	Wii Fit induced significant improvements for the single leg stance test and grip strength only
Adverse events: No	
Esposito et al. (2013), Italia	Children with migraine without aura (F/M, N = 32/39)
Mean age: 9 ± 2 years	Intervention
Two groups: patients vs. healthy controls, F/M, N = 44/49)
Duration: 12 weeks (30 min, three sessions/week)
Location: Home	A choice of 18 balance oriented Wii Fit Plus activities (e.g., Balance Bubble, Hula Hoop, Obstacle Course, Penguin Slide, Rhythm Activities, Segway Circuit, Snowboard Slalom, Skateboard Arena, Ski Jump, Ski Slalom, Soccer Heading, Table Tilt, Tilt City…)	Motor coordination (MABC), fine visuomotricity (Berry-VMI)	Three-month Wii Fit training in children with migraine without aura improved all parameters: motor coordination (including balance skills), and fine visuomotricity. Wii Fit could be used in this population to counterbalance associated developmental delays or other deleterious effects
Adverse events: No	
Ferguson et al. (2013), South Africa	Children with developmental coordination disorders (F/M, N = 22/24)
Age range: 6–10 years
Mean age: 8 ± 1 years	Intervention
Two groups: Wii Fit vs. established neuromotor task training
Duration: six weeks (30 min, three sessions/week)
Location: School	A total of 18 of the Wii Fit games mimicking the act of cycling, soccer, skateboarding and skiing or played with the hand controller	Motor coordination (MABC), physical fitness (functional strength, strength measured with dynamometer, muscle power sprint test, 20 m shuttle run test, adherence	Motor performance improved in the two groups, but more important changes were noted in the traditional training group. The latter was also true for functional strength and cardiorespiratory fitness measurements. Adherence was near 100% in both groups. The choice of one or the other intervention may depend on resources and time constraints
Adverse events: No	
Hammond et al. (2014), UK	Children with developmental coordination disorders (F/M, N = 4/14)
Age range: 7–10 years
Mean age: 8 ± 1 years	Intervention (cross-over RCT)
Two groups: Wii Fit-regular motor training vs. regular motor training-Wii Fit
Duration: four weeks (three sessions/week)
Location: School	A selection of nine Wii Fit games focusing on coordination and balance	Childs: motor proficiency (BOT), self-perceived ability and satisfaction with motor task (coordination skills questionnaire)
Parents: emotional and behavioral development (strengths and difficulties questionnaire)	Wii Fit induced significant gains in motor proficiency and other outcomes for many, but not all the children. Including Wii Fit therapy for children with developmental disorders could be considered
Adverse events: No	
Mombarg, Jelsma & Hartman (2013), The Netherlands	Children with balance alterations (F/M, N = 6/23)
Age range: 7–12 years
Mean age: 10 ± 1 years	Intervention (RCT)
Two groups: Wii Fit vs. passive control
Duration: six weeks (30 min, three sessions/week)
Location: School	A total of 18 activities identified as balance games (mainly from the balance and training plus categories: Ski Jump, Ski Slalom, Snowboard Slalom, Table Tilt, Obstacle Course, Segway Circuit, Skateboard Arena, Tilt City, Rhythm activities…)	Balance (MABC and BOT)	Significant improvements in the Wii Fit group. Effective intervention for children with poor motor development
Adverse events: No	
Jelsma et al. (2014), The Netherlands	Children with probable developmental coordination disorders and balance problems (F/M = 10/18)
Age range: 71–136 months
Mean age: 100 ± 15 months	Intervention
Two groups: Wii Fit vs. waiting period-Wii Fit
Duration: six weeks (30 min, three sessions/week)
Location: Laboratory (supposedly)	A total of 18 “balancing activities” from the Wii Fit Plus software (not including Ski Slalom, which was used for test)	Motor coordination (MABC), balance (BOT, Ski Slalom), enjoyment (home-made scale)	A Wii Fit intervention significantly increased motor and balance skills in children with coordination disorders and balance problems. After six weeks of intervention, 20 children (out of 28) still rated Wii Fit as “super fun” and four as “fun”
Adverse events: No	
Children with cerebral palsy	
Ramstrand & Lygnegård (2012), Sweden	Children with hemiplegic or diplegic cerebral palsy (F/M, N = 10/8)
Age range: 8–17 years
Mean age: 13 ± 3 years	Intervention (cross-over RCT)
Two groups: Wii Fit-no intervention vs. no intervention-Wii Fit
Duration: five weeks (>30 min, five sessions/week)
Location: Home	Balance: Balance Bubble, Ski Jump, Ski Slalom, Soccer Heading, Table Tilt, Tightrope Walk	Primary: balance (modified SOT, reactive balance test, weight shift test)
Secondary: adherence (playing time)	No improvements after the Wii Fit intervention period. A 30 min home-based Wii Fit intervention was not effective to improve balance in children with cerebral palsy. Four children did not complete the required 30 min/day sessions
Adverse events: No	
Jelsma et al. (2013), South Africa	Children with spastic hemiplegic cerebral palsy (F/M, N = 6/8)
Age range: 7–14 years
Mean age: 11 ± 2 years	Intervention (A-B)
One group (cf. details in the paper)
Duration: three weeks (25 min, four sessions/week) + follow-up after two months
Location: Rehabilitation center	Aerobics: Hula Hoop; balance: Balance Bubble, Penguin Slide, Snowboard Slalom, Slalom Ski, Soccer Heading	Balance (BOT), functional mobility (BOT and TUDS)	Balance score improved significantly (sustained at two months follow-up), but not the functional scores (BOT and TUDS). Ten children only preferred to play Wii Fit instead of conventional physiotherapy. Wii Fit may not be used in place of conventional therapy
Adverse events: No	
Tarakci et al. (2013), Turkey	Children with ambulatory cerebral palsy (F/M, N = 3/11)
Mean age: 12 ± 3 years	Intervention
One group
Duration: 12 weeks (40 min, two sessions/week)
Location: Rehabilitation center	Balance: Ski Slalom, Soccer Heading, Table Tilt, Tightrope Walk	Balance (single leg stance test,1 functional reach test, TUG, 6 min walk test)	Balance improved significantly (all outcomes)
Adverse events: No
TUG (sec):
Wii Fit group
Pre-intervention: 18.26 ± 8.95
Post-intervention: 14.57 ± 5.39	
Other populations with balance impairments	
Meldrum et al. (2015), Ireland	Patients with unilateral peripheral vestibular loss (F/M, N = 27/44)
Mean age: 54 ± 15 years	Intervention (RCT)
Two groups: Wii Fit vs. traditional balance rehabilitation
Duration: six weeks (15 min, five sessions/week) + follow-up after six months
Location: Home	Yoga: Deep Breathing, Palm Tree, Standing Knee, Tree Pose; strength: Sideways Leg Lift, Single Leg Extension; balance: Balance Bubble, Ski Slalom, Heading Soccer, Penguin Slide, Table Tilt; aerobics: Advanced Step, Basic Step, Free Step; training plus: Skateboard, Snowball Fight, Table Tilt Plus	Primary: Gait (self-preferred gait speed)
Secondary: gait parameters (various tests performed eyes open or close, including DGI), balance (ABC, SOT), dynamic visual acuity (computerized dynamic visual acuity system), self-perceived benefit (vestibular rehabilitation benefit questionnaire), mental health (Hospital Anxiety and depression Scale), adherence (diary)	Both the Wii Fit and traditional balance training induced improvement in gait speed. No difference was noted between the two groups for gait parameters and other outcomes at the end of the intervention. Adherence was high for the two interventions but Wii Fit was described as more enjoyable
Adverse events: low back pain (N = 1)
ABC (no unit):
Wii Fit group
Pre-intervention: 64.82 ± 18.74
Post-intervention: 74.36 ± 21.25	
Notes:

ABC, activities-specific balance confidence scale; BBS, Berg balance scale; BMI, body mass index; BOT, Bruininks-Oseretsky test; ADL, activities of daily living scale; CHAMPS, community healthy activities model program for seniors; CBM, community balance and mobility scale; COPD, chronic obstructive pulmonary disease; Beery-VMI, Beery visual-motor integration test; CRQ-SR, chronic respiratory questionnaire; CTSIB, clinical test of sensory interaction and balance; DGI, dynamic gait index; EE, energy expenditure; ESWT, endurance shuttle walk test; FAAB, foot and ankle ability measure; FAB, Fullerton advanced balance scale; FES, falls efficacy scale; FIM, functional independence measure; FSS, flow state scale; GDS, geriatric depression scale; HR, hear rate (beats/min); IMI, Intrinsic Motivation Inventory; LASAPAQ, LASA physical activity questionnaire; LEFS, lower extremity functional scale; LLFDI, late life function and disability index; LOS, limits of stability; MABC, movement assessment battery for children; MCTSIB, modified clinical test for sensory interaction in balance; MDRT, multidirectional reach test; METs, metabolic equivalent; MFIS, modified fatigue impact scale; MSIS-29, 29-item multiple sclerosis impact scale; MSWS-12, 12-items multiple sclerosis walking scale; MVC, maximal voluntary contraction of leg extensors; MVPA, moderate-to-vigorous physical activity; NPRS, numeric pain rating scale; OEE, Outcome expectations for exercise scale; PA, physical activity; PACES, physical activity and exercise questionnaire; PAID, problem areas in diabetes scale; PDQ-39, 39-question Parkinson’s disease questionnaire; RFD, rate of force development; RPP, rate pressure product; SE, Schwab & England daily living activities scales; SEE, self-efficacy exercise scale; SEES, subjective exercise experience scale; SF-8, short form-8 health survey; SF-36, short form-36 health survey; SFT, senior fitness test; SOT, sensory organization test; STREAM, stroke rehabilitation assessment of movement; STST, sit-to-stand-test; TD2M, type 2 diabetes mellitus; TUDS, time up and down stairs; TUG, time up and go; UPDRS, unified rating scale for Parkinson’s disease; UTAUT, unified theory of acceptance and use of technology questionnaire; VPA, vigorous physical activity; VO2, oxygen consumption; VO2max, maximal oxygen consumption; WHO-5, five-item WHO well-being index; WHODAS, world health organization disability assessment schedule.

For the same test, unit may vary from one paper to another.

1 Many different single leg stance tests were used in the Wii Fit literature for balance assessment purposes. In this table “single leg stance test” describe any test requiring subjects to stand on one leg.

2 Many different sit-to-stand tests (STST) were used in the Wii Fit literature for balance, strength or functional assessment purpose. In this table, “STST” describes any test that requires the subject to sit and stand repeatedly.

3 Esculier et al. (2012) and Esculier, Vaudrin & Tremblay (2014) report results obtained with the same group of subjects during the same trial.

Goal 3: meta-analyses

The effects of Wii Fit were quantified for selected health-related domains. The most recurrent outcomes noted were the activities-specific balance confidence test (ABC), Berg balance score (BBS) and the time-up-and-go test (TUG). These three tests are frequently used to assess patients’ balance abilities (Powell & Myers, 1995; Berg, 1989; Podsiadlo & Richardson, 1991). ABC is usually administered by a health care professional asking “How confident are you that you will not lose your balance or become unsteady when you…” for 16 different situations (e.g., “…walk around the house?,” “…walk up or down stairs”…). For each item, the participant should answer by expressing confidence in percentage (Powell & Myers, 1995). BBS is a scale able to measure balance in adults. The therapist asks participants to complete 14 different tasks (e.g., “sitting to standing,” “turning to look behind them”…) and evaluates each of them using a five-point score, ranging from 0 to 4 (Berg, 1989). TUG is a simple measure of the time taken by a subject to stand up from a chair, walk a distance of 3 m, turn, walk back to the chair, and sit down (time is expressed in seconds) (Podsiadlo & Richardson, 1991). Firstly, pre- and post-intervention meta-analyses were performed for each of these three outcomes. Secondly, Wii Fit vs. traditional therapy meta-analyses were completed, which only included results from randomized control (RCT) or two-arm trials. The exclusion criteria applied at this stage are described in Table 1. Only studies that used the 3 m version of the TUG test were included. Groups submitted to a combination of Wii Fit activities and more traditional therapy exercises were excluded from the pre- and post-intervention meta-analysis (Barcala et al., 2013; Daniel, 2012; Yatar & Yildirim, 2015). The pre- and post-intervention effect was calculated for the three selected outcomes. These meta-analyses used the mean difference between the reported pre-intervention and post-intervention values. For the Wii Fit vs. traditional therapy meta-analyses, the difference between the pre- and post-Wii Fit intervention changes and the pre- and post-traditional intervention changes were used as inputs in the meta-analysis. The variance imputation methods described by Follmann et al. (1992) were used to estimate the standard deviations of effect size when the authors did not report them. Heterogeneity between studies was assessed using the homogeneity test. A fixed-effect model was used when the I2 statistic, which is the index of heterogeneity, was under 75%. Sub-analyses were conducted in patients and healthy subjects. For ABC, because only two studies included a comparison between Wii Fit and traditional therapy (Yatar & Yildirim, 2015; Meldrum et al., 2015), only the pre- and post-intervention meta-analysis was performed. The risk of bias in each individual study included in the Wii Fit vs. traditional therapy meta-analysis was also assessed (Fig. 2). Meta-regression analyses were performed to assess the impact of intervention duration and volume (i.e., session duration × number of session) on ABC, BBS and TUG. p < 0.05 indicates statistical significance. Meta-analysis was performed using STATA 12.1 (StataCorp, College Station, TX, USA).

Figure 2 Assessment of risk of bias in individual studies included in the Wii Fit vs. traditional therapy meta-analyses.

The absence of ABC, BBS or TUG excluded de facto the studies from the meta-analyses. Therefore the usually reported “reporting bias” was not included in this assessment. No “other bias” was identified.

Results

The literature search provided a total of 279 references of interest (Fig. 1). Following the title and abstract screening process 138 studies were discarded, as they did not meet the selection criteria. One article was not accessible so was also discarded at this stage. An additional 25 references were removed after reading the full-text. Finally, 115 studies were included in the qualitative analysis, covering an approximately six-year period from July 2009 to June 2015.

Goal 1: health domains and populations of interest

The 115 selected studies focused on Wii Fit as a novel tool to improve physical function, fitness or health status. The content of the 115 articles was used to determine the different health domains in which Wii Fit may have potential benefits (Table 2).

Goal 2: systematic review of Wii-Fit interventions

From the 115 selected Wii Fit articles, 68 were intervention studies and met the selection criteria for inclusion in the systematic qualitative review. Overall, these studies involved 2,183 participants from both sexes (females: 1,161, males: 844, not specified: 178), with a wide age range (49 ± 6 months to 86 ± 6 years (Salem et al., 2012; Chao et al., 2013)), and various medical conditions. Primary and secondary outcomes, intervention content, as well as observation period vary from study to study. The intervention durations vary from 2 to 20 weeks (Bower et al., 2014; Toulotte, Toursel & Olivier, 2012), frequencies from 1 to 7 sessions per week (respectively, Toulotte, Toursel & Olivier, 2012; Chan et al., 2012 and Tripette et al., 2014b; Kempf & Martin, 2013) and session time from 10 to 60 min (respectively, Janssen, Tange & Arends, 2013; Franco et al., 2012 and Baltaci et al., 2013; Brichetto et al., 2013; Toulotte, Toursel & Olivier, 2012).

Six papers reported adverse effects: In young adults, light to moderate adverse effects (muscle soreness, pain, sprain, etc.) were observed (Tripette et al., 2014a). Among seniors, hip strain, neck strain, lower back pain as well as one fall were reported (Nicholson et al., 2015; Williams et al., 2010; Agmon et al., 2011). In multiple sclerosis patients, knee pain and lower back pain were also reported (Prosperini et al., 2013). Bower et al. (2014) observed a relatively high rate of falls in stroke patients (four events over a group of 30 patients).

Table 3 describes the characteristics and main results from studies with a primary focus on the effects of Wii Fit interventions on physical activity level, physical fitness or patients’ health status. Among 13 studies, 10 observed positive effects (Janssen, Tange & Arends, 2013; Daniel, 2012; Tripette et al., 2014b; Cutter et al., 2014; Albores et al., 2013; Kempf & Martin, 2013; Cho & Sohng, 2014; Kim et al., 2014; Hoffman et al., 2013, 2014; Chan et al., 2012) and three presented more contrasted results (Owens et al., 2011; Nitz et al., 2010; Albores et al., 2013). Interestingly, four intervention studies were conducted in patients with chronic diseases. They all reported a significant improvement in health status and well-being (chronic obstructive pulmonary disease, type two diabetes mellitus, chronic kidney disease and lower back pain) (Albores et al., 2013; Kempf & Martin, 2013; Cho & Sohng, 2014; Kim et al., 2014). Two reports described Wii Fit interventions as both feasible and effective methods for improving the overall physical fitness, mobility and independence of senior subjects (Janssen, Tange & Arends, 2013; Daniel, 2012; Chan et al., 2012).

For each population, Table 4 summarizes study characteristics and the main results for protocols with a primary focus on the effect of Wii Fit intervention on balance activities and related physical functions. Overall, the qualitative review of these studies supports a positive effect of Wii Fit interventions on balance outcomes. Among the 55 studies, 50 observed a positive effect of Wii Fit on at least one parameter (Morone et al., 2014; Wall et al., 2015; Bieryla & Dold, 2013; Rendon et al., 2012; Bainbridge et al., 2011; Fung et al., 2012; Salem et al., 2012; Barcala et al., 2013; Yatar & Yildirim, 2015; Meldrum et al., 2015; Ferguson et al., 2013; Hammond et al., 2014; Mombarg, Jelsma & Hartman, 2013; Jelsma et al., 2013, 2014; Esposito et al., 2013; Tarakci et al., 2013; Gioftsidou et al., 2013; Melong & Keats, 2013; Lee, Lee & Park, 2014; Cone, Levy & Goble, 2015; Baltaci et al., 2013; Sims et al., 2013; Punt et al., 2015; Brichetto et al., 2013; Nilsagård, Forsberg & von Koch, 2013; Prosperini et al., 2013; Robinson et al., 2015; Hung et al., 2014; Omiyale, Crowell & Madhavan, 2015; Williams et al., 2011; Bateni, 2012; Orsega-Smith et al., 2012; Toulotte, Toursel & Olivier, 2012; Chao et al., 2013, 2014; Cho, Hwangbo & Shin, 2014; Nicholson et al., 2015; Roopchand-Martin et al., 2015; Williams et al., 2010; Agmon et al., 2011; Jorgensen et al., 2013; Esculier et al., 2012; dos Santos Mendes et al., 2012; Padala et al., 2012; Pompeu et al., 2012; Mhatre et al., 2013; Esculier, Vaudrin & Tremblay, 2014; Goncalves et al., 2014; Liao et al., 2015). There were numerous examples were balance-related parameters improved to a similar or even higher extent when using Wii Fit compared to traditional therapies (Morone et al., 2014; Fung et al., 2012; Barcala et al., 2013; Daniel, 2012; Yatar & Yildirim, 2015; Meldrum et al., 2015; Gioftsidou et al., 2013; Melong & Keats, 2013; Lee, Lee & Park, 2014; Baltaci et al., 2013; Sims et al., 2013; Punt et al., 2015; Brichetto et al., 2013; Duclos et al., 2012; Hung et al., 2014; Bateni, 2012; Toulotte, Toursel & Olivier, 2012; Nicholson et al., 2015; Jorgensen et al., 2013; Pompeu et al., 2012; Liao et al., 2015). Only five papers described contrasted results or expressed some reservations about the ability of the software to induce benefits in balance skills (Janssen, Tange & Arends, 2013; Ramstrand & Lygnegård, 2012; Naumann et al., 2015; Bower et al., 2014; Franco et al., 2012), with three of these studies being conducted in healthy populations (Janssen, Tange & Arends, 2013; Naumann et al., 2015; Franco et al., 2012).

Goal 3: outcomes of meta-analyses

For the pre- and post-intervention meta-analyses, seven groups out of six studies were included for ABC, 13 groups out of 12 studies for BBS, and 12 groups out of 12 studies for TUG. For the Wii Fit vs. traditional therapy meta-analyses, 14 groups out of seven studies for BBS, and 12 groups out of six studies for TUG. Studies included in the different meta-analyses involved 595 participants from both sexes (females: 332, males: 242, not specified: 21), with a wide age range (12 ± 3 to 86 ± 5 years (Tarakci et al., 2013; Chao et al., 2013)) and various medical conditions. Whilst these papers all included a measure of ABC, BBS or TUG, the interventions content and duration vary from study to study. The assessment of individual studies revealed a low risk of bias (Fig. 2). Detailed results for ABC, BBS and TUG are presented in Figs. 3–5, and data included in the meta-analyses appears in Tables 3 and 4. Wii Fit interventions did not induce any change in ABC (2.02, 95% CI: −4.01–8.04). For BBS, significant improvements were noted in both healthy subjects and patients (2.00, 95% CI: 0.41–3.60 and 2.99, 95% CI: 0.08–5.90, respectively; 2.23, 95% CI: 0.84–3.63, overall). In addition, there was no significant difference in changes induced by traditional training and those induced by Wii Fit, suggesting that Wii Fit was as valid as traditional training. Regarding TUG, no significant reduction was noted after the Wii Fit intervention in either healthy subjects or patients (−0.34 s, 95% CI: −1.38 to 0.70 and −2.24 s, 95% CI: −5.17 to 0.69, respectively; −0.55 s, 95% CI: −1.53 to 0.43, overall). However, compared to traditional training programs, the Wii Fit did induced a more significant reduction in TUG, especially in patients (−1.76, 95% CI [−2.13 to −1.39], in patients; −1.31, 95% CI [−1.62 to −1.01], overall). The sets of studies included in both BBS and TUG pre- and post-intervention meta-analyses were statistically homogenous (I2 = 0.0%, p = 0.961 and I2 = 0.0%, p = 0.969, respectively for the overall analysis). Various levels of heterogeneity were observed in the Wii Fit vs. traditional therapy meta-analyses (I2 = 60.0%, p = 0.040 for BBS in patients and I2 = 74.3%, p = 0.002 for TUG overall (Figs. 4 and 5), indicating some inconsistencies in the literature. This was expected, however, since different populations were included in the analyses.

Figure 3 Pre- and post-intervention meta-analytic effect for the activities-specific balance confidence test (ABC).

The black point shows the average change for each study. The diamonds describe the pooled values respectively for the change in healthy subjects, patients and the overall population. The vertical black line refers to no change. For each analysis (overall population) or sub-analysis (healthy subjects or patients), a significant effect is observed if the diamond does not touch the black line. The horizontal black line shows the 95% CI and the gray square shows the study weight in percentage. Four-week Wii Fit intervention group (a) and eight-week Wii Fit intervention group (b) (Orsega-Smith et al., 2012). I2: index of heterogeneity.

Figure 4 Pre- and post-intervention meta-analytic effect (A) and Wii-Fit vs. traditional therapy meta-analytic effect (B) for the Berg balance score (BBS).

(A) The black point shows the average change for each study. The diamonds describe the pooled values respectively for the change in healthy subjects, patients and the overall population. The vertical black line refers to no change. For each analysis (overall population) or sub-analysis (healthy subjects or patients), a significant effect is observed if the diamond does not touch the black line. (B) The black point shows the difference of effect between Wii Fit and traditional therapy for each study. The diamonds describe the pooled values respectively for the difference of effect in healthy subjects, patients and the overall population. The vertical black line refers to no difference between Wii Fit-induced change and traditional therapy-induced change. For each analysis (overall population) or sub-analysis (healthy subjects or patients), a significant difference is observed if the diamond does not touch the black line. (A and B) The horizontal black line shows the 95% CI and the gray square shows the study weight in percentage. Four-week Wii Fit intervention group (a) and eight-week Wii Fit intervention group (b) (Orsega-Smith et al., 2012). I2: index of heterogeneity.

Figure 5 Pre- and post-intervention meta-analytic effect (A) and Wii-Fit vs. traditional therapy meta-analytic effect (B) for the time-up-and-go test (TUG).

(A) The black point shows the average change for each study. The diamonds describe the pooled values respectively for the change in healthy subjects, patients and the overall population. The vertical black line refers to no change. For each analysis (overall population) or sub-analysis (healthy subjects or patients), a significant effect is observed if the diamond does not touch the black line. (B) The black point shows the difference of effect between Wii Fit and traditional therapy for each study. The diamonds describe the pooled values respectively for the difference of effect in healthy subjects, patients and the overall population, the vertical black line refers to no difference between Wii Fit-induced change and traditional therapy-induced change. For each analysis (overall population) or sub-analysis (healthy subjects or patients), a significant difference is observed if the diamond does not touch the black line. (A and B) The horizontal black line shows the 95% CI and the gray square shows the study weight in percentage. I2: index of heterogeneity. Unlike ABC and BBS, which are scores, the TUG test results are expressed in time. A negative difference therefore indicates a higher performance.

Meta-regression analyses revealed no significant results (not shown), suggesting no relationships between improvements in balance outcomes and intervention duration or volume.

Discussion

The three main goals set for this review were as follows: Goal 1: Identify the health-related domains in which the Wii Fit series has been tested or used. A scientific database search was undertaken with reasoned exclusion criteria. We identified that the Wii Fit has been used for numerous health purposes and in various populations (Table 2). Balance training was identified as being the most recurrent topic in the literature and appears to be the field of predilection for the usage of the Wii Fit software. Another notable focus was the prevention of metabolic disorders as well as the improvement of health status in people with chronic disease.

Goal 2: Understand the effect of Wii Fit in the identified populations (cf. Goal 1). A qualitative systematic review of studies including Wii Fit interventions was performed, with particular attention given to health and physical activity outcomes. Wii Fit was employed to prevent falls, to induce functional improvements in seniors or in subjects presenting neurodegenerative diseases, to treat orthopedic populations, etc. (Table 4). Overall, the effects of using Wii Fit were mainly positive, with the software being recurrently described as being able to induce similar benefits to traditional therapies. In addition, Wii-Fit interventions were linked to an improvement of health status in several different patients types (diabetic subjects, cancer patients …), however its preventive effect remains to be demonstrated.

Goal 3: To conduct meta-analyses when possible to quantify the effect Wii Fit had on selected health-related domains. In regards to balance training, the results of meta-analyses revealed that Wii Fit interventions had a positive impact on BBS and TUG. Interestingly, Wii Fit interventions also appear very safe, with very low levels of injuries being reported.

Wii Fit for the prevention of metabolic disorders and health status improvement in patients

From light physical activity to moderate-to-vigorous physical activity, AVG elicit a wide range of intensities (Graves et al., 2010; O’Donovan, Roche & Hussey, 2014; Deutsch et al., 2011; Douris et al., 2012; Garn et al., 2012; Lyons et al., 2012; O’Donovan & Hussey, 2012; Tripette et al., 2014a, 2014b; Xian et al., 2014; Worley, Rogers & Kraemer, 2011; Guderian et al., 2010; Mullins et al., 2012; Peng, Lin & Crouse, 2011). However, it is difficult to state whether playing Wii Fit on a regular basis would allow one to meet the American College of Sports Medicine’s recommendations for physical activity or could induce beneficial effects on health. Intervention studies reviewed in this article indicate that playing Wii Fit is not a strategy to consider in young adults (and children) for the prevention of cardio-metabolic disease, because it does not induce any significant increase in physical activity or any improvement in physical fitness (Owens et al., 2011; Nitz et al., 2010). However, one study showed a significant and rapid weight loss during a Wii Fit intervention in postpartum women (Tripette et al., 2014b). Wii Fit may also be a promising tool to aid seniors in maintaining a healthy lifestyle. Intervention studies have reported an increase in physical activity (Janssen, Tange & Arends, 2013), physical fitness (Daniel, 2012) and functional skills (Chan et al., 2012). Playing Wii Fit also clearly appeared to be beneficial for various types of patients: Some studies have reported improvements to health status in chronic obstructive pulmonary disease, hemodialysis patients, diabetic subjects and cancer patients (Albores et al., 2013; Kempf & Martin, 2013; Cho & Sohng, 2014; Hoffman et al., 2013, 2014). While the preventive effects of Wii Fit remain to be demonstrated, the software may be of value in other clinical settings.

Wii Fit for balance training

Many of the intervention studies (55/68) were related to balance training or to the improvement of related functions, with a large majority of them (50/55, Table 4) describing a beneficial effect. The meta-analytic results supported these promising observations. Significant improvements were observed for BBS in both healthy subjects and patients, while a trend was noted for TUG improvements in patients. Interestingly, the meta-analyses also revealed no difference in improvements induced by traditional therapies and Wii Fit interventions for BBS, while TUG showed greater improvements following the Wii Fit intervention compared to after traditional therapy. Taken together, these outcomes suggest a possible therapeutic application for the software, with Wii Fit potentially being as valid as traditional training in some situations. However, a careful look at the qualitative analysis outcome (Table 4) mitigates the overall positive impression for some populations. For instance, Wii Fit intervention outcomes in children with cerebral palsy appeared somewhat contrasted, sometimes being successful (Tarakci et al., 2013), sometimes unsuccessful (Ramstrand & Lygnegård, 2012) or sometimes inducing improvements in some but not all of the parameters (Jelsma et al., 2013). In addition, BBS evaluates balance in isolated balance-related tasks, and TUG combines a limited set of very simple actions (standing-up, walking and sitting-down). Wii Fit-induced improvements were only observed in BBS score and TUG, in a clinical setting, and were not associated with improvements in self-confidence in balance abilities (no changes in ABC), therefore it is unclear whether these improvements can be transferred to activities that occur during daily-life and positively impact the quality of life. The general impressions about Wii Fit interventions are, however, currently positive. Our review should therefore encourage further research in order to assist physiotherapists and health professionals in their decision to incorporate the use of Wii Fit into their treatment regimes. Considering that contrasted observations do exist, prescribing Wii Fit should still be considered with caution.

Wii Fit therapeutic content and Wii Balance Board

It is unsurprising that Wii Fit has been the object of much attention among physical therapists. The specificities of the Wii Fit games taken together with the technical features of the Wii Balance Board tend to promote medial–lateral and anterior–posterior movements, mimicking exercises that are commonly used in physical rehabilitation programs (Levac et al., 2010; Michalski et al., 2012; Duclos et al., 2012). The board is composed of multiple pressure sensors able to work together to follow the displacement of the vertical projection of the center of gravity on the floor. Moreover, the device has been validated against the “gold standard” laboratory-grade force platform for assessing standing balance (Clark et al., 2010). In addition, high levels of adherence have frequently been reported in the reviewed studies (Tables 3 and 4). One may therefore hypothesize that key features of the Wii Fit are the ludic elements that promote adherence in individuals who are not interested in traditional training programs. However, Deutsch et al. (2011) emphasizes one limitation of the Wii Fit, which favors the “knowledge of results” rather than the “knowledge of performance” model, i.e., subjects focus on scores rather than on the quality of movements. This is an important finding, since this would limit the relevance of using Wii Fit at home without the supervision of a therapist checking the quality of movement. One difference with the traditional proprioceptive rehabilitation material is that the Wii Balance Board is unable to tilt. Medial–lateral and anterior–posterior displacements are the result of exteroceptive adaptive mechanisms triggered by visual and auditory feedback stimuli that depend on the game scenario.

Limitations

Firstly, the sub-analyses performed in patients included various pathologies. This was highlighted by the high level of heterogeneity between studies in the Wii Fit vs. traditional therapy meta-analyses (see I2 in Figs. 4B and 5B). While the overall meta-analyses described a positive effect, the results cannot be predictive of Wii Fit intervention-related changes in one specific population. This emphasizes the requirement for more research in order to determine the optimum usage of Wii Fit for each medical domain. Secondly, the attention given to AVG and other virtual reality devices for the purpose of promoting health has been constantly growing, even after the screening period of this review (July 2009 to June 2015). Therefore we encourage readers to also review the new literature on the subject.

Conclusion

Originally designed as a ludic health and fitness promotion software, the Wii Fit series grabbed the attention of physical therapists due to the panel of features favoring body movements. Initial promising observations encouraged physicians, from various medical fields, to test the Wii Fit software on numerous populations. The literature still remains contrasted on the preventive effects of Wii Fit on chronic diseases. However, Wii Fit interventions were shown to be effective for the improvement of health status in various types of patients (chronic obstructive pulmonary disease, hemodialysis, renal complications, diabetes, cancer, etc.). Our review identified that the most notable focus of Wii Fit interventions were balance training. The Wii Fit has indeed been successfully used to prevent falls or to induce functional improvements in a wide range of healthy or pathologic populations (e.g., seniors, subjects with neurodegenerative diseases, orthopedic patients, children with developmental delay, multiple sclerosis patients, etc.). Our meta-analysis supports the general positive impressions about Wii Fit, suggesting promising applications in a wide range of medical fields. The unexpected entry of a video game into the health device market could create innovative healthcare strategies, however, more research is required to validate these claims.

Supplemental Information

Supplemental Information 1 PRISMA flow diagram.

Click here for additional data file.

Supplemental Information 2 PRISMA checklist.

Click here for additional data file.

Additional Information and Declarations

Competing Interests

Author Contributions

Data Availability

The authors declare that they have no competing interests.

Julien Tripette conceived and designed the experiments, performed the experiments, analyzed the data, wrote the paper, prepared figures and/or tables, reviewed drafts of the paper.

Haruka Murakami performed the experiments, analyzed the data, prepared figures and/or tables, reviewed drafts of the paper.

Katie Rose Ryan performed the experiments, wrote the paper, reviewed drafts of the paper.

Yuji Ohta contributed reagents/materials/analysis tools, reviewed drafts of the paper.

Motohiko Miyachi conceived and designed the experiments, contributed reagents/materials/analysis tools, reviewed drafts of the paper.

The following information was supplied regarding data availability:

The work consists of a systematic review and a meta-analysis. No original data have been generated. The data used for the meta-analysis are reported in the tables in the manuscript.

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
