# Peer review of "The contribution of Nintendo Wii Fit series in the field of health: a systematic review and meta-analysis"

_PeerJ, doi:10.7717/peerj.3600_

## Round 0.1 · original submission · Minor Revisions

· Academic Editor

Minor Revisions

The two reviewers and I have been most impressed with the quality of this manuscript. They have both highlighted a number of small areas on which is manuscript can be improved, including the removal of some typographical errors as well as the reorganisation of some the material in the discussion section. If these corrections can be made, we'll fill in this manuscript will add substantially to this evolving literature on the benefits of the Wii Fit for improving health outcomes.

·

Basic reporting

This paper systematically reviews the scientific literature on the use of the Wii Fit for rehabilitation and health promotion purposes. It sums up the areas in which the Wii Fit is currently used and its efficacy in those fields, particularly with respect to balance training. These balance training capabilities are assessed in a meta-analysis of some common outcomes pertaining to balance.

The authors found 115 papers matching their inclusion criteria, spanning a range of fields from obesity-interventions to cognitive rehabilitation for the elderly. They present their systematic review with tabulated paper-summaries categorised by field. This review is very thorough and represents a treasure trove for scientists working in this field.


With respect to basic reporting, in general the paper is well written and draws on the appropriate literature. However there are a couple of minor grammatical errors that will no doubt be picked up given a thorough re-read. For example, on lines 297 & 300 the present tense is used while the past tense is used for the rest of the paragraph. On lines 161/162 “have been” should be “were”, etc.

Secondly, this review effectively contains three different reviews of three different sets/subsets of papers, as shown in Figure 1. While the results section makes this division apparent, the discussion does not. I would recommend the authors slightly rework the discussion such that the three goals of the review are addressed independently before a more general discussion. This may be as simple as adjusting the existing subheadings to make it clearer exactly which set of papers are being discussed when.

The meta-analysis of ABC is rather brushed over around line 237, but I think it is worth including, particularly because a forest-plot contains much more information than simply a pooled effect size estimate. Given the mention of the ABC meta-analysis in the abstract, and the equal attention given to it in Table 4, it seems incomplete to leave the forest-plot missing from the paper.

On a related note, it is rather confusing that the horizontal axis changes direction between Figure 3 and Figure 4. In Figure 3, positive evidence for the Wii Fit is on the right hand side of the forest plot, whereas in Figure 4 it is on the left hand side of the plot. I would recommend that one of these figures be reversed for consistency.

Finally, three measures are meta-analysed: the ABC, BBS and TUG. Although all these tests are used to assess balance capabilities, the authors include little description of them. They are first mentioned around line 108, where the authors do reference a handful of papers with more complete explanations. However, given the prominence of these three measures in the paper, and the fact that this paper does not overtly assume knowledge of balance training, I suggest readers might benefit from a short description of each.
I do commend the authors for the sentences around line ~339 where they compare these measures in the context of the results of the meta-analyses.

Experimental design

The date range of articles searched for this systematic review is from “winter 2009 to June 2015”. Firstly, which month is meant by “winter” should be clarified, and secondly June 2015 was a long time ago, which naturally limits the relevance of this review. However, I am aware that the size of this field has grown considerably in the past two years, and bringing this systematic review up to date is likely infeasible within a reasonable time frame. Furthermore, the likely size of such an updated review would make for difficult reading/interpretation. As such my recommendation is that the authors simply state clearly that the datedness of their review is a limitation.

This aside, the methods are clearly written and sensible, and I am pleased with the inclusion of a PRISMA checklist and flow diagram.

Validity of the findings

I feel that the discussion, which has room for expansion if the authors desired, is suitably supported by the results of the meta-analyses and the systematic review.

Reviewer 2 ·

Basic reporting

This review summarizes the contribution of Nintendo Wii Fit series for health and fitness. Meta-analyses were performed and the results show that Wii Fit can sufficiently be used as a rehabilitation tool in various clinical cases. The literature review of this paper is impressive. The analytical method shown in this review is quite decent. The writing and organization of the paper are good. Therefore, the reviewer do not hesitate to recommend that this paper is suitable for PeerJ after the authors revising the paper according to the following comments:

Minor Points:
1. Table 2 only cites the references without details, which is not informative. More details about the cited references shown in Table 2 should be provided.
2. The Conclusion part is not informative. More analytical results should be shown in the Conclusion.
3. Several typos should be corrected. Please double-check the manuscript.

Experimental design

Since it is a review paper, there is no necessary part for experimental design here.

Validity of the findings

The statistical method looks quite suitable. The conclusions and claims are generally well-supported by the analysis.

---

## Round 0.2 · accepted · Accept

· Academic Editor

Accept

The reviewers and I are happy with the changes you've made to the manuscript and would like to notify you that the paper has now been accepted for publication in PeerJ.

·

Basic reporting

No comment

Experimental design

No comment

Validity of the findings

No comment

Additional comments

In short, I am satisfied with the rebuttal letter and revisions made by the authors. Many typographical errors have been fixed, missing diagrams and measure-descriptions have been added, and the discussion has been reformatted to summarise the results more clearly.

Although not ideal, the paragraph at the end of the discussion detailing the age of the articles in this review is sufficient.

I am happy to recommend that this article be accepted.

Reviewer 2 ·

Basic reporting

This revised version has addressed all my previous concerns. Now I recommend that this paper in its current form is acceptable to PeerJ.

Experimental design

no comment

Validity of the findings

no comment